# A small set of conserved genes, including *sp5* and *Hox*, are activated by Wnt signaling in the posterior of planarians and acoels

Aneesha G. Tewari[1,2], Jared H. Owen[1,2], Christian P. Petersen[1¤a], Daniel E. Wagner[1,2¤b], Peter W. Reddien[1,2,3]*

**1** Whitehead Institute for Biomedical Research, Cambridge, Massachusetts, United States of America, **2** Department of Biology, Massachusetts Institute of Technology, Cambridge, Massachusetts, United States of America, **3** Howard Hughes Medical Institute, Chevy Chase, Maryland, United States of America

¤a Current address: Department of Molecular Biosciences, Northwestern University, Evanston, United States of America
¤b Current address: Department of Systems Biology, Harvard University, Boston, Massachusetts, United States of America
* reddien@wi.mit.edu

**Data Availability Statement:** All RNA-sequencing files were deposited to the NCBI sequence read archive under BioProject PRJNA558985. Gene

## Abstract

Wnt signaling regulates primary body axis formation across the Metazoa, with high Wnt signaling specifying posterior identity. Whether a common Wnt-driven transcriptional program accomplishes this broad role is poorly understood. We identified genes acutely affected after Wnt signaling inhibition in the posterior of two regenerative species, the planarian *Schmidtea mediterranea* and the acoel *Hofstenia miamia*, which are separated by >550 million years of evolution. Wnt signaling was found to maintain positional information in muscle and regional gene expression in multiple differentiated cell types. *sp5*, *Hox* genes, and Wnt pathway components are down-regulated rapidly after *β-catenin* RNAi in both species. *Brachyury*, a vertebrate Wnt target, also displays Wnt-dependent expression in *Hofstenia*. *sp5* inhibits trunk gene expression in the tail of planarians and acoels, promoting separate tail-trunk body domains. A planarian posterior *Hox* gene, *Post-2d*, promotes normal tail regeneration. We propose that common regulation of a small gene set–*Hox*, *sp5*, and *Brachyury*–might underlie the widespread utilization of Wnt signaling in primary axis patterning across the Bilateria.

## Author summary

How animals form and maintain their body axes is a fundamental topic in developmental biology. Wnt signaling is an important regulator of head-tail axis formation across animals, with high Wnt signaling specifying tail identity. In this study, we use two species that are separated by more than 550 million years of evolution, planarians and acoels, to find genes regulated by Wnt signaling in the tail broadly in the Bilateria. We identified a small conserved set of Wnt-regulated genes, including the transcription factor-encoding genes *sp5* and *Hox*. This suggests that regulation of this gene set might be a key function

sequences were submitted to GenBank with accession numbers MN275808-MN275830, MN305295-MN305319, MN381936-MN381938 and MN400983.

**Funding:** P.W.R. is funded by a grant from the NIH (R01GM080639). P.W.R. is funded by the Howard Hughes Medical Institute. P.W.R. is funded by the Eleanor Schwartz Charitable Foundation A.G.T. was funded by the National Science Foundation Graduate Research Fellowship Program The funders had no role in study design, data collection and analysis, decision to publish, or preparation of the manuscript.

**Competing interests:** The authors have declared that no competing interests exist.

of Wnt signaling in the tails of bilaterally symmetric animals. Inhibition of a planarian posterior *Hox* gene, *Post-2d*, by RNAi caused tail-regeneration defects. Inhibition of *sp5* by RNAi revealed that it functions to restrict the expression of trunk genes in the tail of planarians and acoels. Since Wnt signaling activates both trunk and tail patterning gene expression in planarians, this suggests a mechanism by which Wnt signaling can establish separate trunk-tail body domains through regulation of *sp5*.

## Introduction

How body axes are formed is a central problem in animal development and evolution. The body plans of bilaterally symmetric animals (the Bilateria) are typically organized along the anterior-posterior (AP) or head-tail axis, the dorsal-ventral (DV) or back-belly axis, and the medial-lateral (ML) axis. Pre-bilaterians, such as cnidarians, can have tissue organized along a primary body axis (oral-aboral). Despite the large diversity of body plans across the Metazoa, a central feature of anterior-posterior and oral-aboral axes is the polarized expression of Wnt signaling ligands that act through the transcriptional effector β-catenin, commonly referred to as canonical Wnt signaling, and their antagonists [1]. This can result in a gradient of Wnt signaling activity that regulates pattern of tissue distribution on the primary axis during development and regeneration [1]. For example, in *Hydra*, Wnt signaling promotes oral identity on the oral-aboral axis, and over-activation of the Wnt pathway leads to formation of ectopic tentacles along the body column [2,1]. In planarians, inhibition of Wnt signaling causes loss of tail identity and the formation of ectopic heads in the posterior and laterally [3–5]. In frogs, a gradient of Wnt signaling activity is required for AP neural patterning [6,7]. In mice, Wnt signaling is required during early AP axis establishment, and its activation marks the future posterior [8,9]. A key question that arises from the observation that Wnts specify posterior fates broadly across animals is whether a conserved Wnt-dependent transcriptional program acts to establish these posterior identities.

Here, we used the freshwater planarian *Schmidtea mediterranea* and the acoel *Hofstenia miamia* to study conserved Wnt-dependent transcriptional changes that inform posterior identity across bilaterians. Acoels are worms within the phylum Xenacoelomorpha that has been placed at the base of the Bilateria, in a sister taxa to all other bilaterians [10–15]. This current phylogenetic placement suggests that planarians and acoels are separated by >550 million years of evolution and that comparisons between them can provide insight into core features of bilaterians [16]. Both species utilize adult stem cells called neoblasts to replace differentiated cells in a process of tissue turnover [17,12,18,19]. Both species require Wnt signaling for maintenance of the AP axis during tissue turnover, as well as for AP regeneration [3–5,20,12]. Although AP axis maintenance and re-establishment are not equivalent to axis formation during embryogenesis, their study in these species can provide insight into conserved patterning mechanisms that also occur during development. Because posterior Wnt signaling activity is constitutive in planarians and acoels, Wnt-regulated genes are continuously expressed in the tail, enabling their study.

In planarians, multiple studies have utilized RNAi of *β-catenin-1* and Wnt ligands to study Wnt signaling and to identify the effects of Wnt pathway inhibition or activation on AP axis pattern formation [3–5,21–30]. RNAi of *β-catenin* genes, which encode the downstream effector of canonical Wnt signaling, leads to progressive loss of tail identity and gain of head identity in planarians and acoels [3–5,20,12,25,30]. Wnt signaling has been found to promote the expression of tail and trunk patterning genes and to inhibit the expression of anterior-

patterning genes [3–5,24,25,27,28,30]. The targets of Wnt signaling in acoels remain poorly understood. We inhibited *β-catenin* genes by RNAi and performed RNA sequencing of tails at early time points post-RNAi in planarians and acoels to identify acutely Wnt-regulated genes in both species. Genes encoding patterning factors, well known to be expressed in planarian muscle, acutely change expression after Wnt inhibition. In addition we found that broadly distributed tissues, such as the epidermis and gland cells, also display regional gene expression in a Wnt-dependent manner. Our findings indicate that Wnt signaling can actively maintain regional gene expression in multiple adult differentiated cell types. A small group of conserved genes were most sensitive to changes in Wnt signaling in both species. This group included *Hox* genes and a gene encoding the transcription factor SP5, which is a direct Wnt target in vertebrates [31,32]. In planarians, these genes were expressed in many tissue types and displayed the earliest detectable gene expression pattern changes during regeneration, after the initial phase of wound-induced gene expression. *Post-2d*, a planarian posterior *Hox* gene, was found to be required for normal tail regeneration. *sp5* is much less studied than *Hox* genes as a candidate central regulator of the AP axis of animals broadly. Recent work in *Hydra* has shown that *sp5* acts in a feedback loop with Wnt3 to restrict head organizer formation [33]. Functional analysis of *sp5* in planarians and acoels revealed that it acts to down-regulate expression of trunk genes in the tail. In planarians, trunk gene expression is Wnt-dependent. This suggests the existence of a Wnt signaling circuit that can promote different spatial identities on the primary axis through activation of SP5 in the tail and SP5-mediated inhibition of Wnt-regulated trunk genes.

## Results

### A posterior program of gene expression is acutely down regulated after *β-catenin-1* inhibition in planarians

In order to identify genes with expression regulated by Wnt signaling in the planarian posterior, we used RNAi to inhibit *β-catenin-1*, which encodes the downstream effector of Wnt signaling. Prior studies have identified a variety of planarian genes that change their expression following *β-catenin-1* perturbation [24,25,27,30]. *ndl-3* and *ptk-7* are known to be down regulated by 12 days post *β-catenin-1* RNAi and *teashirt* is down regulated as early as 4 days after *β-catenin-1* inhibition [24,27]. RNA sequencing studies found a host of genes that display expression changes by 16 days following initiation of *β-catenin-1* RNAi, a timepoint at which 788 genes were differentially expressed [25]. To identify the genes most acutely regulated by Wnt signaling, we utilized tails from uninjured animals and RNA sequencing at early timepoints (day 1, day 2, day 4, and day 6) following RNAi initiation (Fig 1A). We reasoned that this early-timepoint post-RNAi approach could determine the changes that occur prior to AP-tissue-type transformation and therefore identify the Wnt-regulated program that lies upstream of tissue-type identity. Differential expression analysis was performed to identify genes down and up regulated at each of these time points with a $p_{adj} < 0.05$ and a $\log_2$-fold change less than -0.5 or greater than 0.5. 1,276 genes (636 down-regulated, 638 up-regulated) that were differentially expressed at any timepoint were subjected to hierarchical clustering to study patterns of gene expression change after *β-catenin-1* inhibition (S1A and S2A Figs and S1 Table). At the earliest timepoint, only 43 genes were differentially expressed in this analysis.

A cluster of 52 genes behaved similarly to *β-catenin-1* itself, rapidly displaying decreasing transcript levels after *β-catenin-1* RNAi, many starting as early as day 1 and the rest by day 2 post-RNAi initiation (Fig 1B and 1C, S1A Fig and S2 Table). We reasoned that genes that follow a similar gene expression trajectory as *β-catenin-1* after *β-catenin-1* RNAi are good candidates to be targets of Wnt signaling. In addition, 29/52 genes in this cluster were contained

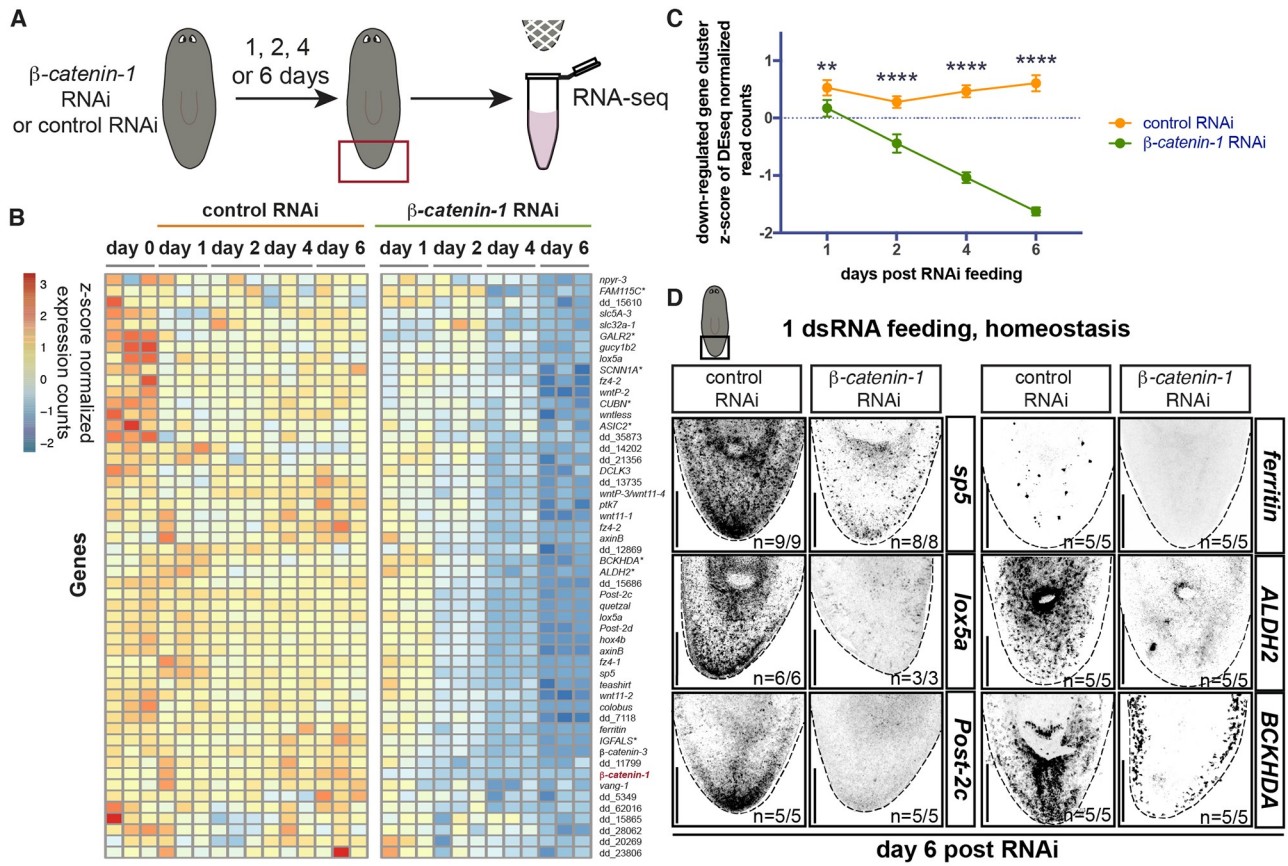

**Fig 1. A posterior program of gene expression is acutely down-regulated after *β-catenin-1* inhibition in planarians.** (A) Experimental scheme for RNA sequencing. *Schmidtea mediterranea* adults were fed once with control or *β-catenin-1* dsRNA. Tails were collected in trizol 1, 2, 4, and 6 days after feeding for RNA sequencing. Red box indicates tissue collected. (B) A cluster of 52 genes is down-regulated early after *β-catenin-1* RNAi. Heatmap displaying gene expression counts as z-scores for indicated timepoints post-RNAi feeding. *indicates annotation by best BLAST hit. Gene list provided in S2 Table. (C) Significant down-regulation of gene expression begins at day 1 post-*β-catenin-1* RNAi. Mean z-score of gene expression counts for 52 genes in 1(B) at indicated time points post-RNAi feeding. Data is plotted as mean ± S.D.**p<0.01 ****p<0.001 (D) Validation of genes down-regulated after *β-catenin-1* RNAi. Fluorescence *in situ* hybridisation (FISH) for indicated transcripts at day 6 post RNAi in homeostasis. Images are presented in grayscale with color inverted. Black box indicates region shown. Scale bars, 200μm.

within the 348 genes previously reported to be significantly down-regulated 16 days after *β-catenin-1* RNAi [25] (S2 Table). Thus, our approach identified a subset of genes regulated by Wnt signaling that respond acutely to *β-catenin-1* RNAi and identified novel genes that were not found in previous analyses. 34/52 genes in this cluster had known expression patterns or were screened here by *in situ* hybridization. 29/34 were expressed in the planarian posterior including 19/22 genes known to be expressed in a posterior-to-anterior transcriptional gradient in planarian muscle cells [34,28] (S2 Table). This set of 19 genes includes posteriorly expressed position control genes (PCGs)–genes with constitutive regional expression that are predicted to be part of a planarian-patterning pathway or have a patterning abnormal RNAi phenotype. We confirmed that expression of posterior genes *sp5*, *lox5a*, and *Post-2c* known to be down regulated by day 16 post-*β-catenin-1* RNAi, are down regulated as early as day 6 [25] (Fig 1D). *Smed-ferritin*, which was expressed in a few cells in the tail, was also down regulated early after *β-catenin-1* RNAi (Fig 1D). The previously uncharacterized *ALDH2*, *BCKHDA*, and *gucy1b2* genes were expressed in the posterior epidermis and additionally in other regions and cell types (S1B–S1F Fig). Using *in situ* hybridization we found that in each case, specifically the

posterior expression domains of these genes were *β-catenin-1* dependent whereas expression in other regions was not reduced 6 days after *β-catenin-1* RNAi (Fig 1D and S1B–S1F Fig). These findings indicate that posterior expression of patterning molecules in muscle and posterior expression of genes in the epidermis acutely require Wnt signaling in the posterior.

We also identified a cluster of 56 genes that were steadily up-regulated in planarian tails after Wnt inhibition (Fig 2A and 2B, S2A Fig and S2 Table). Significant up-regulation of these genes, which include known anterior PCGs, began by day 2 post initiation of RNAi and was robust by day 4 post-RNAi (Fig 2C). This indicates that the up-regulation of patterning molecules associated with head identity after Wnt inhibition occurs much earlier than previously known, but still following the initial down-regulation of posterior genes. 34/56 genes identified in this cluster were among 440 genes previously determined to be significantly up-regulated 16 days after *β-catenin-1* RNAi [25] (S2 Table). 30/56 genes had known expression patterns, or were screened here by *in situ* hybridization. 28 of these 30 genes were expressed in the planarian anterior, including 14/21 genes previously identified to be regionally expressed in anterior muscle [28] (S2 Table). *in situ* hybridizations revealed that uncharacterized genes in this cluster were anteriorly expressed, with several expressed in the epidermis and parenchymal cell types (Fig 2D and S2B–S2I Fig). For instance, dd_8729 was expressed in the eyes and head tip. After *β-catenin-1* RNAi this gene was ectopically expressed in foci in the posterior (Fig 2D). The appearance of ectopic expression foci resembled changes in the expression of anterior PCGs, such as *sFRP-1* (encoding a candidate Wnt-inhibitory secreted frizzled-related protein), after *β-catenin-1* inhibition, with foci marking the location of ectopic head formation [3,25,28]. The genes dd_12049, dd_6380, and dd_11499 were expressed in the anterior epidermis, with expression spreading toward the posterior after *β-catenin-1* RNAi (Fig 2D and S2B–S2E Fig). dd_5811 and dd_1379 were expressed in the pre-pharyngeal region in parenchymal cell types and dd_2184 (named, *trunk regionally expressed in gland cells 1*, *trig-1*) was expressed in *mag-1$^+$* marginal adhesive glands cells in the trunk, with little detectable expression in the tail (Fig 2D and S2F–S2I Fig). However, the cell types in which these genes are expressed do exist in the tail [35], suggesting that the dd_5811, dd_1379, and *trig-1* genes are regionally expressed within parenchymal cell type populations that exist broadly on the AP axis. After *β-catenin-1* RNAi, expression of these genes increased in the posterior parenchyma (Fig 2D). This suggests that in addition to ectopic foci of anterior gene expression in the posterior, inhibition of Wnt signaling can lead to shifting domains of gene expression along the AP axis. Whereas AP regional expression of genes in planarian muscle is well established, our data suggests that other cell types, such as the epidermis, marginal adhesive gland cells, and other parenchymal cell types, also regionally express genes along the AP axis. The anterior restriction of such regional expression in multiple differentiated tissues can be maintained by constitutive Wnt signaling.

### Wnt-dependent maintenance of posterior gene expression and restriction of posterior boundaries of anterior gene expression can occur dynamically in differentiated tissues

Tissue turnover occurs constitutively in planarians and involves new cell production by a stem cell population called neoblasts [17,19]. Previous work has shown that ectopic expression of anterior PCGs like *sFRP-1* in the tail of planarians after *β-catenin-1* inhibition requires formation of new tissue [25]. To determine whether early changes in gene expression after *β-catenin-1* inhibition could occur in existing mature cells or required new cells generated from neoblasts during tissue turnover, we utilized irradiation to ablate neoblasts and RNA sequencing to study this question at a transcriptome-wide scale (S3A Fig). 10/52 genes described above to be acutely down-regulated after Wnt inhibition displayed significantly reduced expression

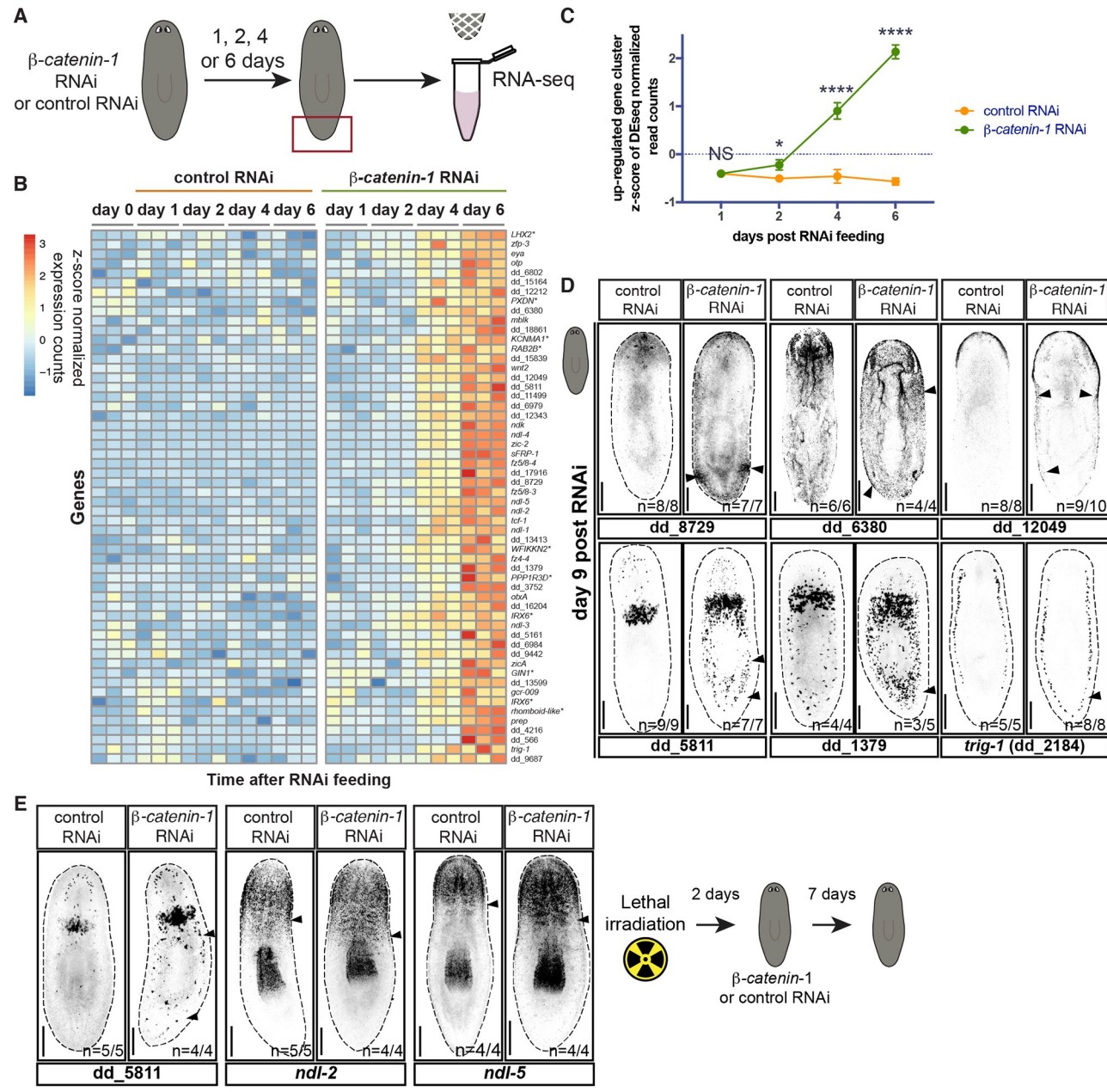

**Fig 2. An anterior program of gene expression is acutely up-regulated after *β-catenin-1* inhibition in planarians.** (A) Experimental scheme for RNA sequencing. *Schmidtea mediterranea* adults were fed once with control or *β-catenin-1* dsRNA. Tails were collected in trizol 1, 2, 4, and 6 days after feeding for RNA sequencing. Red box indicates tissue collected. (B) 56 genes are up-regulated early after *β-catenin-1* RNAi in planarian tails. Heatmap displaying gene expression counts as z-scores for indicated timepoints post-RNAi feeding. *indicates annotation by best BLAST hit. Gene list provided in S2 Table. (C) Up-regulation of gene expression begins by day 2 post-*β-catenin-1* RNAi. Mean z-score of gene expression counts is plotted for 56 genes up-regulated in 2(B) at indicated time points post-RNAi feeding. Data is plotted as mean ± S.D. *p<0.05 ****p<0.001 (D) Validation of genes up-regulated after *β-catenin-1* RNAi. FISH for indicated transcripts at day 9 post RNAi in unamputated animals. Images are presented in grayscale with color inverted. (E) Posterior expansion of anterior expression domains after *β-catenin-1* RNAi is irradiation insensitive. FISH for dd_5811, *ndl-2*, and *ndl-5* after control and *β-catenin-1* RNAi in irradiated animals during homeostasis. Arrows indicate posterior boundary of gene expression. Images are presented in grayscale with color inverted. Scale bars, 200μm.

by day 4 post-*β-catenin-1* RNAi in irradiated animals as well (S3B Fig and S3 Table). This list included genes encoding Wnt-pathway components, such as *fz4-1* (encoding a Frizzled Wnt-receptor-family protein), and a gene encoding the transcription factor SP5 (S3B Fig). Genes expressed in non-muscle differentiated cells in the posterior (*BCKHDA* and *ferritin*) also displayed reduced expression. Although 42/52 genes in this subset did not pass a significance threshold of $p_{adj} < 0.05$, many still displayed reduced expression after *β-catenin-1* RNAi in tails of irradiated animals. We validated the reduction of several of these genes by *in situ* hybridization. *sp5*, as well as the *Hox* genes *hox4b* and *lox5a* displayed reduced expression after *β-catenin-1* RNAi in irradiated animals, suggesting that down-regulation of posterior gene expression can occur in existing tissue, instead of only passively by the replacement of posterior cells with cells of anterior identity (S3C Fig).

Previous work has shown that formation of ectopic foci of anterior PCG expression in the posterior after *β-catenin-1* RNAi requires the formation of new tissue [25]. RNA-sequencing revealed that 1/56 genes previously identified as up-regulated early after *β-catenin-1* RNAi (dd_5811) displayed significantly increased expression in the tail after irradiation as well (S3D Fig and S3 Table). This result was validated by *in situ* hybridization (Fig 2E). Therefore, the posterior expansion of dd_5811 expression in the parenchyma does not require new cell formation. Although no anterior PCGs passed a significance threshold of $p_{adj} < 0.05$, *wnt2* displayed increased expression after irradiation in *β-catenin-1* RNAi tails by RNA-sequencing (S3D Fig). Indeed, the normally anterior-restricted expression domain of *wnt2* expanded posteriorly after *β-catenin-1* RNAi in irradiated animals by *in situ* hybridization (S3E Fig). Previous work has shown that anterior PCGs, such as *ndk*, *ndl-1*, *ndl-2*, and *ndl-5* also display subtle posterior expansion of anterior expression domains after *β-catenin-1* RNAi [28,30]. For *ndl-2* and *ndl-5*, this expansion occurs before formation of ectopic foci of PCG expression in the posterior [28]. We found that posterior expansion of *ndl-2* and *ndl-5* gene expression also occurred in irradiated *β-catenin-1* RNAi animals (Fig 2E). These findings suggest that Wnt-dependent maintenance of posterior gene expression and restriction of anterior gene expression domains, for at least some regionally expressed genes occurs actively in differentiated cells. However, fate changes in neoblasts appear to mediate the subsequent changes in regional gene expression that lead to ectopic foci of anterior PCG expression and head formation.

## A posterior program of gene expression is down-regulated after *β-catenin-1* RNAi in acoels

Acoels are worms within the phylum Xenacoelomorpha that represent a sister taxa to all other bilaterians [10–15]. This phylogenetic position allows acoels to serve as an outgroup to the major clades of the Bilateria. Processes that are similar between acoels and one or more other clades of bilaterians are good candidates to have been present in the last common ancestor of the Bilateria, and to be widespread in extant animals. The acoel *Hofstenia miamia* has emerged as a new and powerful experimental model system [12,36]. Inhibition of Wnt signaling in *Hofstenia miamia* by *β-catenin-1* RNAi is known to cause AP-patterning defects during regeneration and homeostasis [12].

We sought to determine which Wnt-sensitive genes in planarians are also Wnt-sensitive in acoels. Wnt-dependent gene expression changes have not been thoroughly studied in acoels. Therefore, we first identified early timepoints after initiation of *β-catenin-1* RNAi at which detectable changes in gene expression occurred in *Hofstenia*. We used *in situ* hybridization to study known PCGs in uninjured animals. Ectopic expression of the anterior PCG *sFRP-1* in the posterior occurred at a low frequency by day 6 post-initiation of *β-catenin-1* RNAi, and in all animals tested by day 14 after RNAi (S4A Fig). We detected reduced expression of the

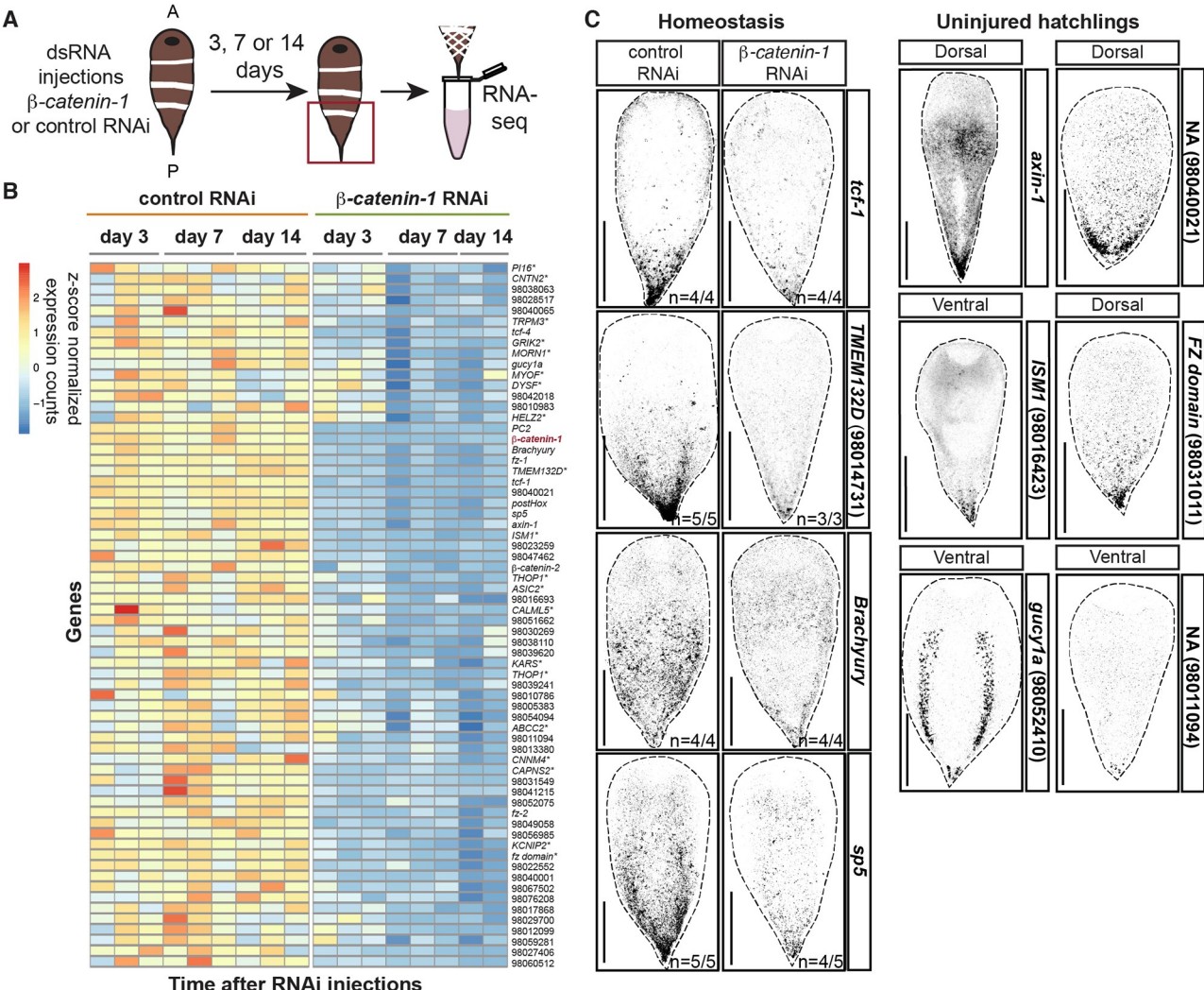

**Fig 3. A posterior program of gene expression is down-regulated after β-catenin-1 inhibition in acoels.** (A) Experimental scheme for RNA sequencing. *Hofstenia miamia* were injected once a day for three consecutive days with control or *β-catenin-1* dsRNA. Tails were collected in trizol at 3, 7, and 14 days after the first injection for RNA sequencing. Red box indicates tissue collected. (B) A cluster of 66 genes is down-regulated early after *β-catenin-1* RNAi. Heatmap displaying gene expression counts as z-scores for indicated timepoints post first RNAi injection. *indicates annotation by best BLAST hit. Gene list provided in S5 Table. (C) Left: Validation of *β-catenin-1* RNAi-dependent down-regulation of gene expression. FISH for indicated transcripts at day 9 post-RNAi initiation in homeostasis. Images are presented in grayscale with color inverted. Right: Genes down-regulated after *β-catenin-1* RNAi are expressed in the posterior. FISH for indicated transcripts in two week old hatchlings. Transcript ID provided in parentheses. Images are presented in grayscale with color inverted. Scale bars, 200μm.

posterior PCG *fz-1* by day 14 post-RNAi initiation (S4A Fig). Based on these results we performed injections of control or *β-catenin-1* dsRNA followed by RNA-sequencing of tails at day 3, day 7, and day 14 post injection, to capture gene expression changes before and after detectable PCG changes occurred by *in situ* hybridization (Fig 3A). Differential expression analysis was performed at each of these time points and all differentially expressed genes with $p_{adj} < 0.1$ and a $\log_2$-fold change less than -0.5 or greater than 0.5 were subjected to hierarchical clustering to study temporal patterns in gene expression (S4B and S4C Fig and S4 Table). A more lenient $p_{adj}$ cutoff was used in *Hofstenia miamia* because of the larger phenotypic variability observed after *β-catenin-1* RNAi compared to planarians.

We found, similar to the case in planarians, a group of 66 genes that were steadily down-regulated after *β-catenin-1* inhibition and that clustered with *β-catenin-1* (Fig 3B, S4B Fig and S5 Table). This cluster contained the known posterior PCGs *fz-1* and *fz-2* [12]. We cloned 13 genes from this cluster and found that 11/13 were expressed in the tail by *in situ* hybridization (Fig 3C). Notable among these genes was an *sp5* transcription factor-encoding gene, a *Hox* gene, a *Brachyury* gene encoding a T-box transcription factor, and genes encoding predicted Wnt-pathway components including *axin-1*, *tcf-1*, and *tcf-4*, as annotated by best human BLAST hit. A group of 92 genes was up-regulated after *β-catenin-1* RNAi (Fig 4A and 4B, S4C Fig and S5 Table). This cluster included the anteriorly expressed genes *wnt-5* and *fz-7* [12]. Candidates from this cluster were screened by *in situ* hybridization to determine their expression patterns. Out of 28 genes cloned, we were able to determine the expression pattern of 11. All 11 genes were expressed in the anterior (Fig 4C). Expression of these genes was found to spread toward the posterior after *β-catenin-1* RNAi (Fig 4C). This set included genes annotated as *ZIC3* and *PKNOX2 (prep)* by best human BLAST hit. Homologs of *zic* and *prep* genes are important regulators of anterior regeneration in planarians [37–39]. These anterior genes were expressed in multiple differentiated cell types including the epidermis and sub-epidermal cells (S4D–S4E Fig).

## A small set of conserved genes are Wnt targets in planarians and acoels

We compared the *β-catenin-1* RNAi RNA-seq datasets obtained from planarians and acoels to identify genes that are candidates to represent a Wnt-regulated posterior program widely in the Bilateria. We compared the group of genes that clustered with *β-catenin-1* after RNAi in planarians and in acoels by BLAST ($e<1^{-06}$), and obtained a list of 5 gene families that were down-regulated in both species. This included an *sp5* family transcription factor gene, *Hox* genes, *frizzled* homologs, an *axin* gene, and a *guanylate cyclase* homolog (Fig 5A). Phylogenetic analysis was used to determine that the *sp* family transcription factor genes identified in both species encode SP5 homologs (S5A Fig). In planarians, of thirteen known *Hox* genes, four were down-regulated early after *β-catenin-1* RNAi. These four genes are expressed in the posterior. *Post-2c* and *Post-2d* have been previously annotated as posterior *Hox* genes with high confidence by phylogenetic analyses [40]. *hox4b* has not been confidently placed by phylogenetic analysis and *lox5a* belongs to the spiralian-specific 'lox' *Hox* group [40]. RNAi has yet to reveal a patterning role for *Hox* genes in planarians [40,28]. In *Hofstenia miamia*, a single posterior *Hox* gene homolog, *Hof-postHox*, was down-regulated early after *β-catenin-1* RNAi (Fig 3B). We identified two additional *Hofstenia miamia Hox* genes: a *Hox1* homolog (*Hof-Hox1*) and a central *Hox* (*Hof-centHox*)(S5B Fig), consistent with analyses of *Hox* genes from other acoels [41,42]. *Hof-Hox1* and *Hof-centHox* were expressed broadly along the anterior and mid-body of juveniles (S5B and S5C Fig). Expression of *Hof-postHox* was not detectable by FISH. We also found a *Hofstenia caudal* homolog (*Hof-Cdx*), which was expressed in the trunk and tail of hatchlings, along with some expression in the anterior (S5B and S5C Fig). *Cdx* was not significantly affected after Wnt inhibition by RNA sequencing. We found that both *guanylate cyclase* homologs identified as down-regulated after Wnt inhibition in planarians and acoels were expressed in the posterior and encode subunits of the GUCY1 family of soluble guanylate cyclase proteins (Fig 3C, S1F and S6A Figs). These proteins can act as receptors for nitric oxide or oxygen [43]. Planarian dd_12650 was found to encode a predicted member of the Gucy1b2 family of beta-subunits, and the encoded protein lacks a heme nitric oxide-binding domain. The *Hofstenia miamia* 9852410 transcript encodes a predicted member of the Gucy1a family of alpha subunits (S6A Fig). Finally, the T-box transcription factor-encoding *Brachyury* gene is a known Wnt target in vertebrates [44,45]. Planarians do not have a detected *Brachyury*

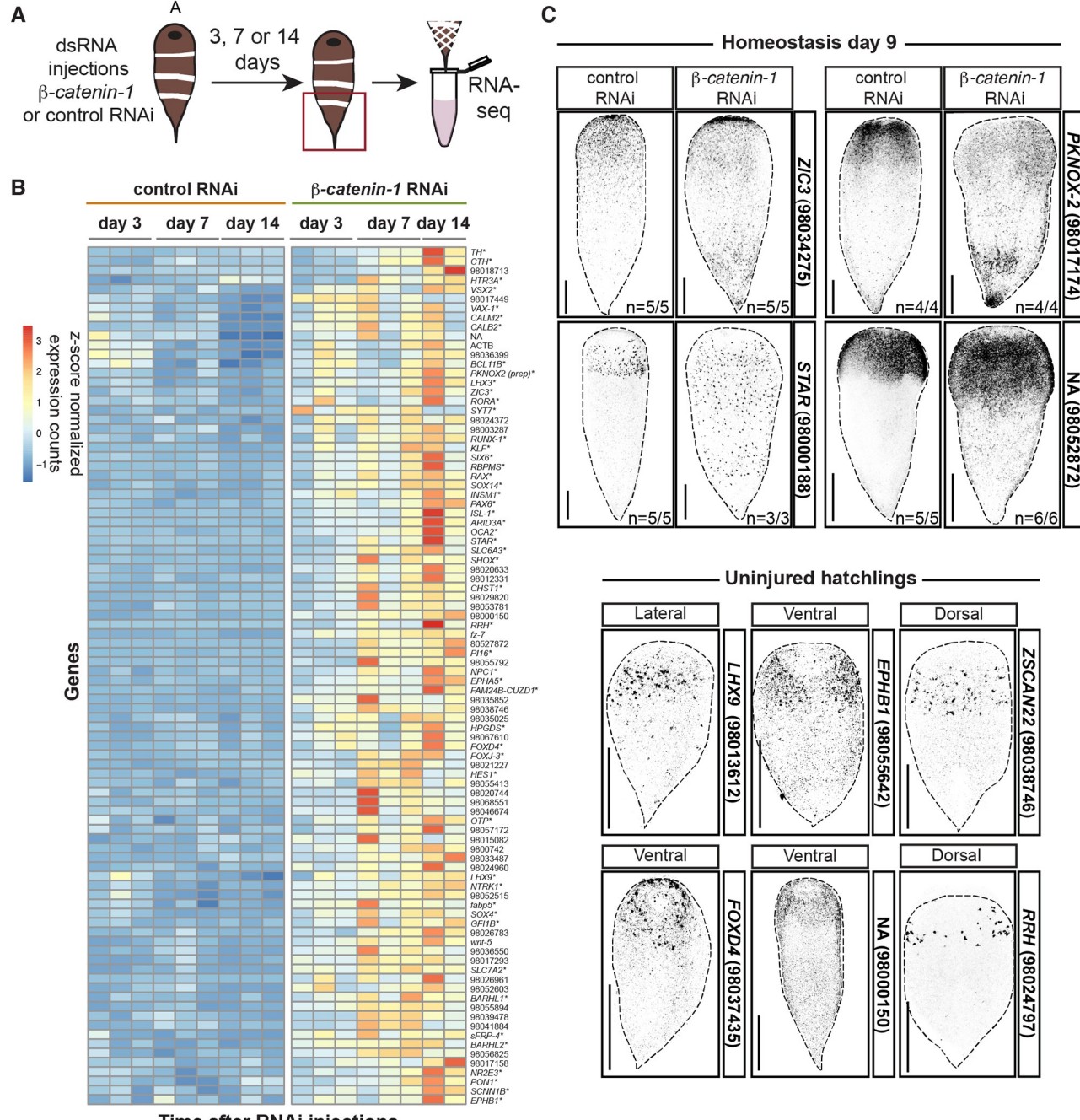

**Fig 4. An anterior program of gene expression is up-regulated after *β-catenin-1* inhibition in acoels.** (A) Experimental scheme for RNA sequencing. *Hofstenia miamia* were injected once a day for three consecutive days with control or *β-catenin-1* dsRNA. Tails were collected in trizol at 3, 7, and 14 days after the first injection for RNA sequencing. Red box indicates tissue collected. (B) 92 genes are up-regulated early after *β-catenin-1* RNAi. Heatmap displaying gene expression counts as a z-score for indicated time-points post-RNAi initiation. *indicates annotation by best BLAST hit. Gene list provided in S5 Table. (C) Top: Validation of *β-catenin-1* RNAi dependent up-regulation of gene expression. FISH for indicated transcripts at day 9 post-RNAi initiation in homeostasis. Images are presented in grayscale with color inverted. Bottom: Genes up-regulated after *β-catenin-1* RNAi are expressed in the anterior. FISH for indicated transcripts in two week old hatchlings. Transcript ID provided in parentheses. Images are presented in grayscale with color inverted. All scale bars, 200μm.

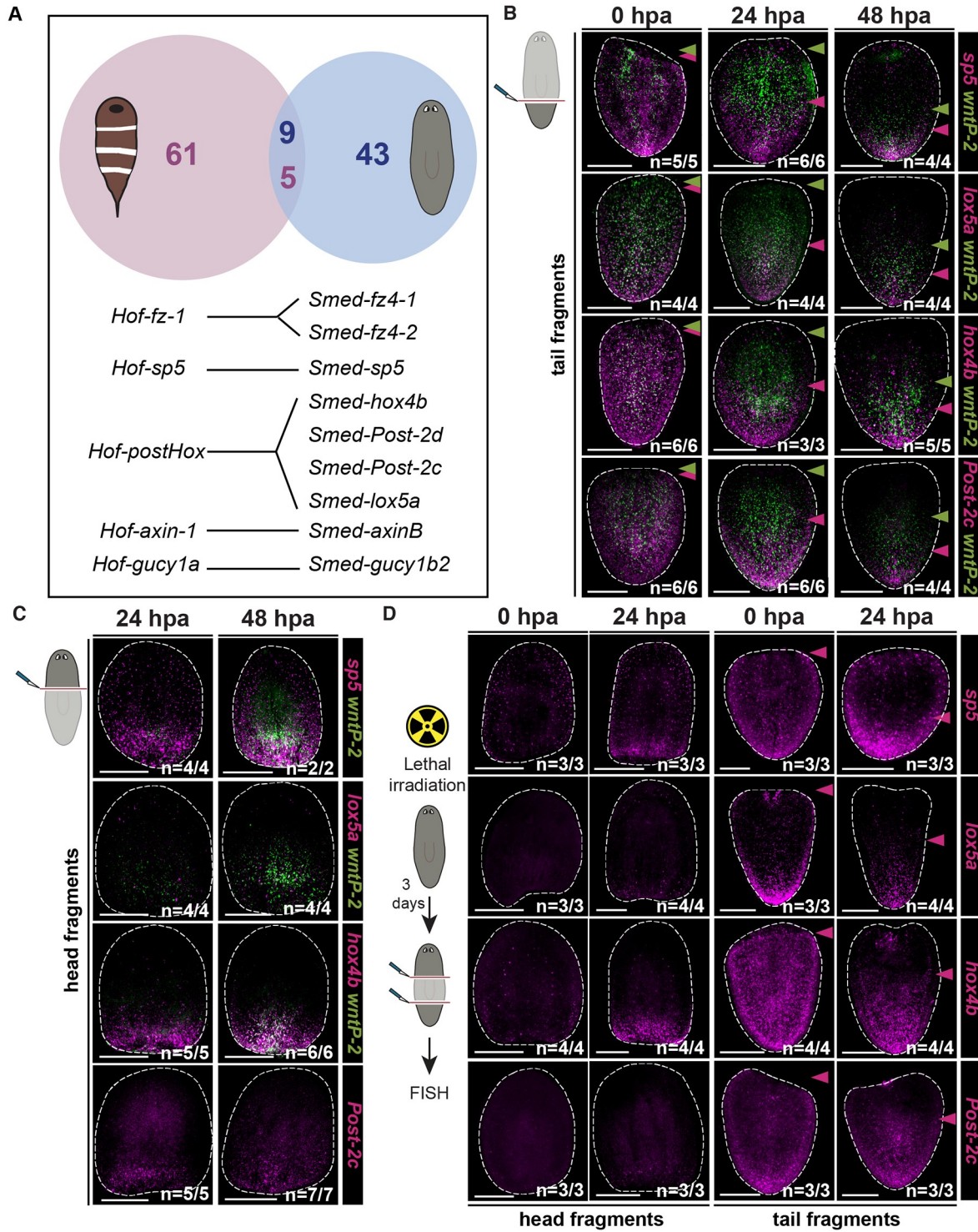

**Fig 5. A small number of conserved genes are regulated by Wnt signaling in both planarians and acoels.** (A) A small group of genes is down-regulated early after *β-catenin-1* RNAi in planarians and acoels. Top: Venn diagram displaying overlap between *Schmidtea mediterranea* and *Hofstenia miamia* down-regulated clusters as determined by BLAST (e value < 1E-06). Bottom: List of genes obtained from BLAST overlap analysis. (B) Expression of planarian *sp5* and *Hox* genes re-scales prior to *wntP-2* re-scaling in regenerating tail fragments. FISH for *sp5*, *lox5a*, *hox4b*, *Post-2c* (magenta), and *wntP-2* (green) at 0, 24, and 48 hours post amputation in regenerating tails. (C) Planarian *sp5* and *Hox* genes are expressed by 24 hours at posterior-facing wounds. FISH for *sp5*, *lox5a*, *hox4b*, *Post-2c* (magenta), and *wntP-2* (green) at 24 and 48 hours post amputation in regenerating heads. (D) Changes in the expression domains of planarian *sp5* and *Hox* genes during regeneration are irradiation insensitive. FISH for *sp5*, *lox5a*, *hox4b*, and *Post-2c* (magenta) after lethal irradiation in regenerating heads (left) and tails (right) at 0 and 24 hours post amputation. Scale bars, 200μm.

homolog [46] (S6B Fig), however, a gene encoding a T-box-family transcription factor was down-regulated early after Wnt inhibition in *Hofstenia miamia*. Phylogenetic analysis revealed that this gene in fact encodes a *Brachyury* homolog and that acoels have 7 additional Tbx family genes (S6B Fig). These findings suggest that a common and small Wnt-driven posterior program of gene expression including *sp5*, posterior *Hox* genes, *Brachyury*, *guanylate cyclase* genes, and genes encoding Wnt-pathway components themselves, is conserved across distantly related extant bilaterians.

### *sp5* and *Hox* gene expression domains are re-established early during regeneration

The small list of conserved Wnt-regulated genes between planarians and *Hofstenia* prominently highlights *Hox* genes and *sp5*. Since Wnt signaling plays a key role in AP axis regeneration, we sought to determine the dynamics of *sp5* and *Hox* gene expression changes after amputation in planarians with *in situ* hybridizations. *sp5* was expressed at posterior-facing wounds of regenerating head and trunk fragments by 24 hours after amputation (S7A Fig). This result is consistent with analysis of Wnt-regulated tail genes, including *sp5*, *Hox*, and posterior PCGs like *wntP-2*, at 16 hour wounds of regenerating fragments by RNA sequencing [30]. In addition, the expression of *sp5* in regenerating trunk and tail fragments was re-scaled by 24 hours post-amputation (S7A Fig). This time-point for re-scaling from anterior-facing wounds precedes not only new tissue differentiation but also substantial changes in the expression of canonical posterior PCGs like *wntP-2* [22,23]. To determine if changes in *sp5* and *Hox* gene expression during regeneration precede re-establishment of known PCG expression domains we performed double FISH for these genes and the posterior PCG *wntP-2*. *lox5a*, *hox4b*, and *Post-2c*, like *sp5*, all displayed re-scaled gene expression in tail fragments by 24 hours post amputation (Fig 5B, S7G Fig). *hox4b* and *Post-2c* displayed robust expression by 24 hours at posterior-facing wounds of regenerating head fragments (Fig 5C). Conversely, *wntP-2* did not display re-scaled expression away from the anterior-facing wound of tail fragments by 24 hours post-amputation and was only lowly expressed by 24 hours post-amputation at the posterior-facing wound of head fragments (Fig 5B and 5C, S7G Fig). In addition, these changes in the expression of *sp5* and *Hox* genes during regeneration were irradiation insensitive (Fig 5D). This suggests that during planarian regeneration, after wound-induced gene expression, conserved, acutely Wnt-sensitive targets like *sp5* and *Hox* are the first genes to change expression domains, preceding changes in other patterning genes and occurring in pre-existing tissue near the injury.

### *sp5* and *Hox* genes are expressed in multiple cell types in the posterior

To determine the identity of cells expressing *sp5*, we examined *sp5* expression in a planarian transcriptome atlas obtained from extensive single-cell RNA sequencing [35]. By plotting only cells sequenced from *S. mediterranea* tails we found that *sp5* is expressed in almost every major tissue type in the planarian posterior (Fig 6A and 6B). By contrast, most planarian posterior PCGs are expressed largely specifically in muscle [34]. We utilized FISH to further assess the identities of posterior cell types that express *sp5*, and found that *sp5* was expressed in neurons, neoblasts, muscle, *cathepsin*$^+$ cells (which include phagocytic cells [47]), and marginal adhesive gland cells in the tail (Fig 6C). Interestingly, *Post-2d* and *Post-2c* also displayed broad expression in multiple cell types in the posterior, whereas *hox4b* and *lox5a* were largely expressed in muscle and epidermis (Fig 6D–6G). Other conserved Wnt targets *axinB*, *fz4-1*, *fz4-2*, and *gucy1b2* were also expressed in multiple cell types in the posterior (S7B–S7F Fig). These data indicate that conserved, acutely Wnt-sensitive genes in the planarian posterior display expression in a diverse range of differentiated tissue types.

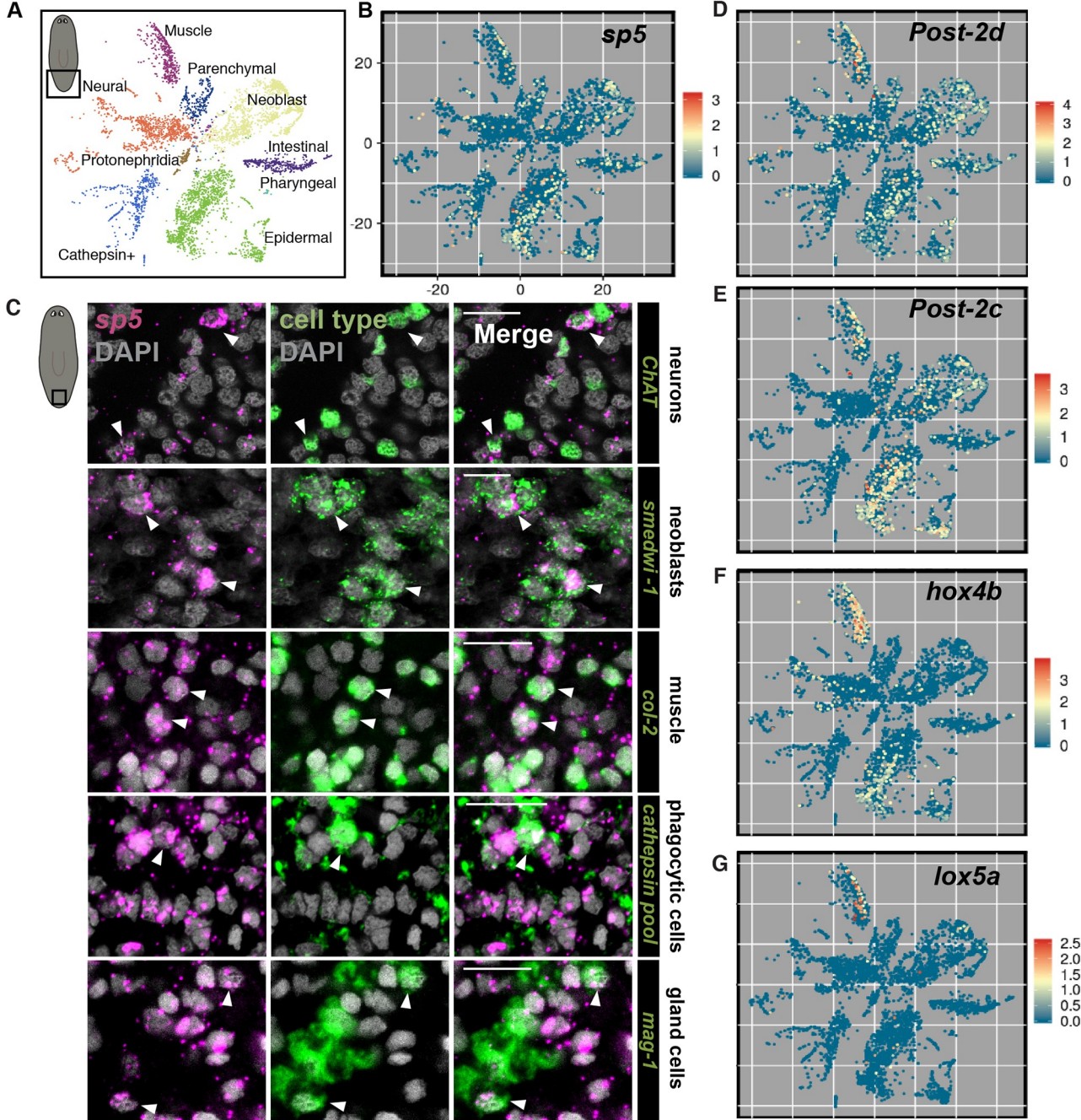

**Fig 6. Planarian *sp5* and *Hox* genes are expressed in multiple cell types in the tail.** (A) t-SNE plot key of single cells from the planarian tail. Each major tissue type is labelled with a color. (B) *sp5* is expressed in multiple cell types in the tail. tSNE-plot of cells obtained from the tail colored by *sp5* gene expression (red, high; blue, low). (C) *sp5* is expressed in all major cell types in the tail. FISH for *sp5* (magenta), differentiated cell type marker (green) and DAPI (nuclear stain, grey) in uninjured planarian tails. *cathepsin* cells are marked by a pool of RNA probes for transcripts; dd_1161_1, *pbgd-1*, *if-1*, *cali*, dd_582 (*CTSL2*), dd_9, dd_10872, dd_1831, dd_5690, *aqp-1*(dd_1103), dd_7593, dd_6149, and dd_1260. (D) *Post-2d* is expressed in multiple cell types in the tail. tSNE-plot of cells obtained from the tail colored by *Post-2d* gene expression (red, high; blue, low). (E) *Post-2c* is expressed in multiple cell types in the tail. tSNE-plot of cells obtained from the tail colored by *Post-2c* gene expression (red, high; blue, low). (F) *hox4b* is expressed in muscle and epidermis in the tail. tSNE-plot of cells obtained from the tail colored by *hox4b* gene expression (red, high; blue, low). (G) *lox5a* is expressed in muscle in the tail. tSNE-plot of cells obtained from the tail colored by *lox5a* gene expression (red, high; blue, low). Scale bars, 10μm.

## *Post-2d* RNAi leads to defects in planarian tail regeneration

*Hox* genes are well-known mediators of AP-axis patterning in bilaterian development. Although *Hox* genes are expressed during regeneration in many organisms, their functional role during this process is poorly understood. The role of planarian *Hox* genes in adult homeostasis and regeneration is unknown despite efforts to inhibit their expression by RNAi [40,28]. We performed RNAi of all four planarian *Hox* genes identified as acutely down-regulated after *β-catenin-1* RNAi in this study (*Post-2c*, *lox5a*, *Post-2d* and *hox4b*). Each gene was inhibited individually and all four genes were inhibited together. No defects were observed in homeostasis or regeneration after RNAi of *Post-2c*, *lox5a*, *hox4b* or combinatorial RNAi of all four genes. RNAi of *Post-2d* caused tail-regeneration defects after amputation in head and trunk fragments (Fig 7A and S8A Fig). Regenerating tails appeared flat or indented at 5 days after amputation (Fig 7A). Head regeneration appeared normal in *Post-2d* RNAi animals (Fig 7A and S8A Fig). During posterior regeneration, *Post-2d* RNAi animals expressed the posterior PCGs *wntP-2*, *wnt11-1*, and *wnt11-2*, and formed a *wnt1*⁺ posterior pole, suggesting posterior patterning occurred (Fig 7B and S8B Fig). However, posterior-pole cells in *Post-2d* RNAi head fragments frequently deviated from the mid-point of the wound compared to control fragments (Fig 7B). In addition, *Post-2d* RNAi tails lacked expression of the epidermal DV-boundary marker NB.22.1e at the location of indentation, demonstrating a defect in production of normal tissues at the regenerating tail tip (Fig 7C). We also inhibited the *Hofstenia postHox* gene and did not observe any overt macroscopic phenotype. RNA sequencing of *postHox* RNAi tails from uninjured animals, however, showed down-regulation of *Brachyury* (S8C Fig).

## *sp5* inhibits trunk gene expression in the tail of planarians and acoels

The role of *sp5* in bilaterian AP-axis patterning, is poorly understood. *sp5* is known to be a direct target of Wnt signaling in multiple vertebrate species [31,32]. SP5 protein is also known to directly interact with Tcf1/Lef1 at β-catenin-target gene enhancers [48]. However, it is debated whether SP5 activates or represses transcription of β-catenin targets [48,49]. Recent work in *Hydra* has shown that *Hy-sp5* is also positively regulated by Wnt signaling and represses Wnt targets [33]. In order to determine the role of *sp5* in planarians, we performed *sp5* RNAi followed by RNA sequencing of tails at various time points in uninjured animals. Analysis of this sequencing data revealed a small number of genes affected after *sp5* RNAi. We did not detect any significant changes in the expression of posterior genes known to be *β-catenin-1* sensitive. However, we found that the expression of two PCGs, *ndl-3* and *ptk-7*, was up-regulated in the tail after *sp5* RNAi at all time points (Fig 8A and 8B and S7 Table). The expression of both of these genes is dependent on *β-catenin-1*, but both of these genes are expressed in the planarian trunk, with expression excluded from the tail [50,25,27] (S8D Fig). This suggests that *sp5* might act to repress the expression of a subset of *β-catenin* targets in the tail. Another previously uncharacterized gene (*trig-1*) was highly up-regulated after *sp5* RNAi (Fig 8A). This gene is expressed in marginal adhesive gland cells in the trunk region of planarians (S2H and S2I Fig). However, after *sp5* RNAi, expression of this gene extended into the tail in *mag-1+* cells (Fig 8B). This gene was also identified as up-regulated after *β-catenin-1* RNAi in tails (Fig 2B and 2D). These data suggest that *sp5* inhibits the expression of multiple Wnt-regulated trunk genes in the tail, promoting subdivision of trunk and tail.

We also performed *sp5* RNAi followed by RNA-sequencing of tails in uninjured *Hofstenia*. We found that the expression of candidate Wnt pathway genes was affected: two *frizzled* genes were up-regulated after *sp5* RNAi in the tail and *sFRP-4* was down-regulated (Fig 8C and S7 Table). In addition, a novel gene *spreg-1* (*sp5 regulated-1*) was normally expressed in the

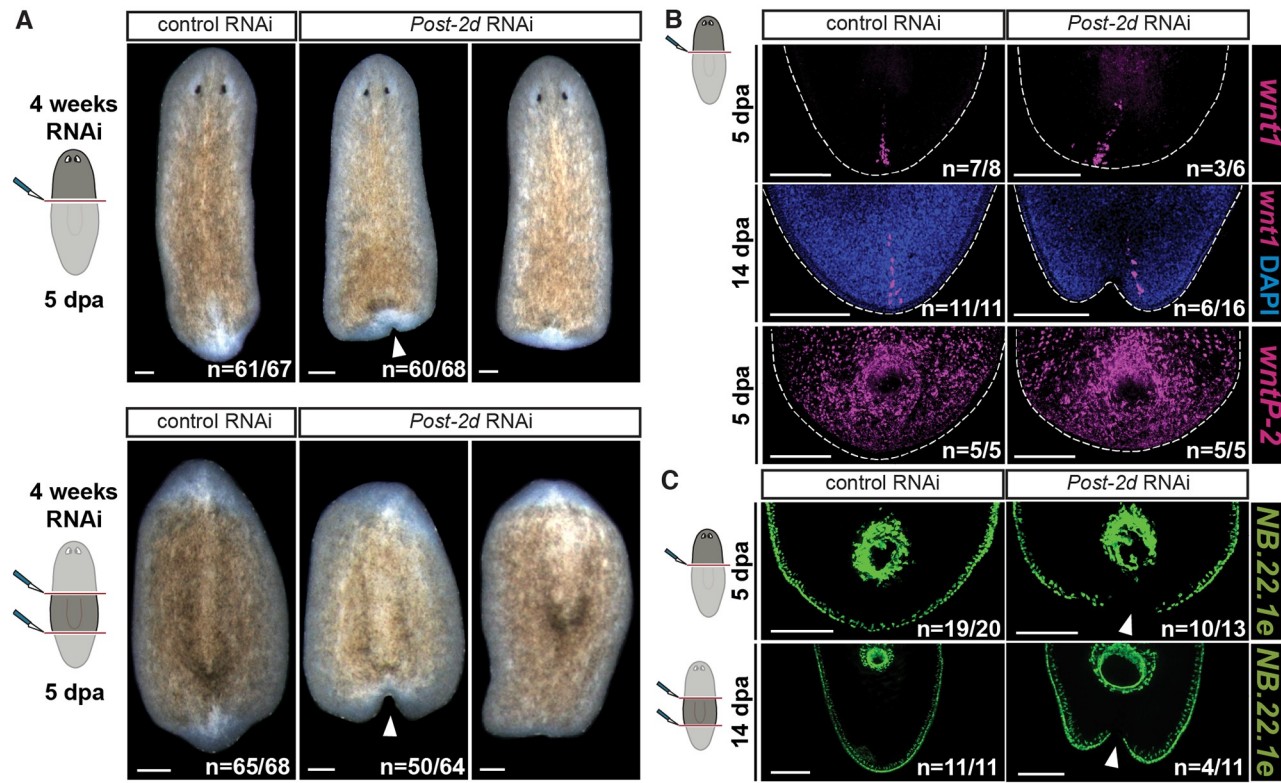

**Fig 7. *Post-2d* RNAi leads to tail regeneration defects in planarians.** (A) *Post-2d* RNAi animals regenerate indented tails after amputation. Live images of regenerating head fragments (top) and trunk fragments (bottom) after indicated RNAi conditions at 5 days post amputation. (B) *Post-2d* RNAi head fragments express posterior-patterning genes during regeneration. FISH for *wnt1* (posterior pole) and *wntP-2* at indicated time points post amputation in regenerating head fragments. (C) *Post-2d* RNAi head fragments fail to express an epidermal DV boundary marker at the posterior tip. FISH for NB.22.1e (DV boundary, mouth) at indicated time points post amputation in regenerating fragments. All scale bars 200μm.

anterior and trunk of *Hofstenia* but not appreciably in the tail (Fig 8D). Expression of this gene spread into the tail after *sp5* RNAi (Fig 8C and 8D). This suggests that acoel *sp5*, like planarian *sp5*, acts to inhibit trunk gene expression in the tail.

## *sp5* functions with *wntP-2* to restrict trunk identity along the planarian AP axis

To determine if *sp5* plays a functional role in maintaining restricted trunk identity we inhibited planarian *sp5* and assessed posterior maintenance and regeneration. We did not detect any abnormalities with *sp5* RNAi alone. To test if *sp5* might work with Wnt ligands to affect trunk identity we performed double RNAi of *sp5* and *wntP-2*. *wntP-2* RNAi is known to cause the formation of ectopic mouths and pharynges posterior to existing structures in the trunk [27,28]. In addition, like *sp5* RNAi, *wntP-2* RNAi leads to posterior expansion of *ndl-3* expression into the tail [27,28]. *sp5* RNAi enhanced the *wntP-2* RNAi phenotype. Compared to *wntP-2*; control RNAi animals, a greater proportion of *sp5; wntP-2* double RNAi animals formed ectopic mouths, and ectopic mouths in this condition appeared more developed (Fig 8E and S8F Fig). This occurred despite comparable inhibition of *wntP-2* expression in both *wntP-2*; control and *sp5; wntP-2* RNAi conditions (S8E Fig). This suggests that *sp5* acts with Wnts to maintain trunk identities in their appropriate location along the AP axis.

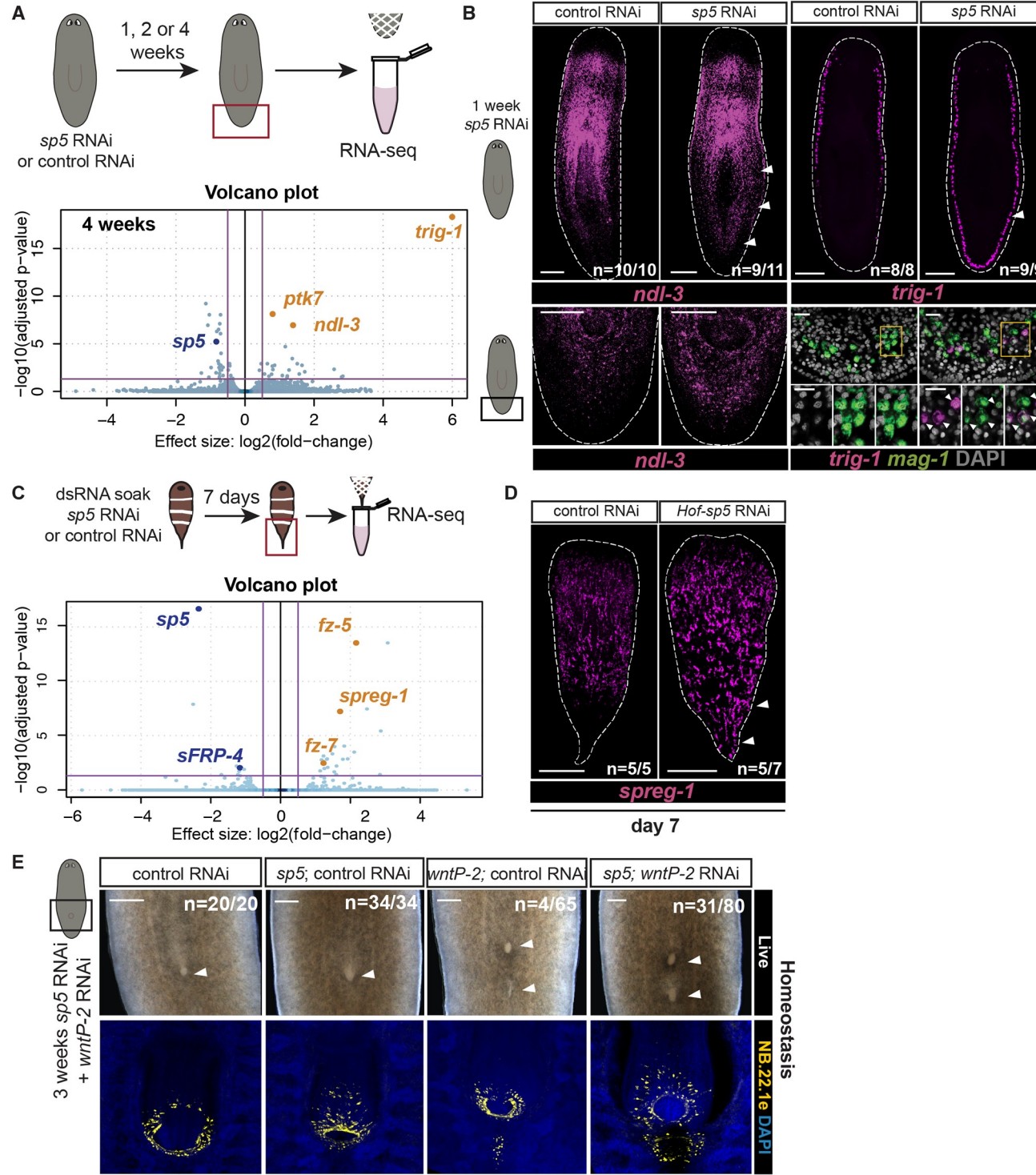

**Fig 8. *sp5* promotes distinction of trunk-tail gene expression domains.** (A) Trunk genes *ndl-3*, *ptk7*, and *trig-1* are up-regulated in the tail after *sp5* RNAi in *Schmidtea*. Top: Experimental scheme for RNA sequencing. Animals were fed with *sp5* dsRNA for 1,2, or 4 weeks followed by collection of tails for RNA-sequencing. Bottom: Volcano plot displaying genes differentially expressed between control and *sp5* RNAi tails at 4 weeks with a $p_{adj}$ <0.05 and $\log_2$-fold change <-0.5 or >0.5, Differential expression analysis provided in S7 Table. (B) Trunk genes *ndl-3* and *trig-1* are up-regulated in the tail after *sp5* RNAi. FISH for *ndl-3* (left; magenta) and *trig-1* (right top; magenta) after 1 week of *sp5* RNAi. Scale bars, 200μm. Ectopic *trig-1* expression (right bottom; magenta) in the tail after *sp5* RNAi occurs in *mag-1*⁺ cells (right bottom; green). Scale bars, 20μm. (C) *Hofstenia sp5* RNAi leads to up-regulation of a novel trunk gene in the tail. Top: Experimental scheme. RNA sequencing on *Hofstenia* tails 7 days after initiation of dsRNA soaking for *sp5* or control RNAi. Bottom: Volcano plot displaying genes differentially expressed between control and *sp5* RNAi tails with a $p_{adj}$ <0.05 and $\log_2$-fold change <-1 or

>1, Differential expression analysis provided in S7 Table. (D) *Hof-sp5* restricts expression of a novel trunk gene in the tail. FISH for *spreg-1* (magenta) in indicated RNAi conditions at day 7 post RNAi initiation in uninjured animals. Scale bars, 200μm. (E) Planarian *sp5* RNAi enhances the formation of ectopic mouths caused by *wntP-2* RNAi. Left: Feeding regimen for RNAi. Right: Live images of the mouth after 3 weeks of RNAi feeding (ventral view). FISH for mouth marker NB.22.1e (yellow). Scale bars 200μm.

## Discussion

Despite the large complexity of body plans that exist across the animal kingdom, there are a small number of signaling pathways that regulate axis formation. Polarized Wnt signaling is a nearly ubiquitous feature of anterior-posterior or oral-aboral axis formation across metazoans [1]. How this signaling pathway can lead to the formation of complex patterns across the head-tail axis is a fundamental question of animal development and evolution. Here we investigated the genes regulated by Wnt signaling in two distantly related bilaterians (planarians and acoels). Our findings (i) identify mechanisms and molecules by which Wnt signaling generates axial pattern, and (ii) identifies a small set of broadly conserved genes that are commonly regulated by Wnt signaling in these species. These two categories of findings are discussed below.

### Patterning mechanisms of the planarian AP axis

In planarians, Wnt signaling regulates anterior-posterior positional information, which is largely harbored in muscle [3–5,34,24,25,27,28,30]. Previous work has shown that this includes promoting posterior and trunk PCG expression and inhibiting anterior PCG expression [3–5,24,25,27,28,30]. These PCG expression domains were acutely regulated by Wnt signaling in our data—i.e., changing, for some PCGs, in as little as 1–2 days following *β-catenin-1* RNAi. The broad PCG expression domains of the AP axis were dynamically regulated by Wnt signaling in mature muscle cells. For instance, *β-catenin-1* RNAi in irradiated animals lacking neoblasts caused reduction in posterior PCG expression and shifting of the posterior boundary of some anterior PCG expression domains (in the pre-pharyngeal region). Wnt inhibition leads to the appearance of foci of anterior PCG gene expression in the posterior, associated with ectopic head formation [3,25,28]. The appearance of these foci in the posterior occurs subsequent to the irradiation-insensitive changes to the global pattern of PCG expression [28] and does not occur following *β-catenin-1* RNAi in irradiated animals [25]. Our findings together with prior work on Wnt signaling in planarians support the model that Wnts act in a morphogen-like manner to maintain regional gene expression in muscle cells along the planarian AP axis.

Previous studies in planarians have found that many genes with expression impacted by Wnt signaling are expressed in muscle, neoblasts, and neoblast progeny [25,30]. Our approach revealed that not only muscle and neoblasts, but broadly distributed cell types such as the epidermis and parenchymal cells can regionally express genes along the AP axis in a Wnt-dependent manner. This raises the possibility that these broadly distributed differentiated cell types play AP regional roles in planarian physiology. Adult axial tissue pattern emerges in most animals from transient, embryo-specific patterning events (e.g., limb bud formation). In adult planarians, prior work suggests that region-appropriate specification and migration of stem cell progeny can also generate and maintain axial tissue pattern. We suggest, based upon findings here, a third process for generating and maintaining tissue pattern in adults: a broadly dispersed field of differentiated cells (such as skin and gland cells) can display region-appropriate gene expression that is actively regulated by constitutive adult positional information.

Another subset of Wnt-regulated genes, including *sp5* and *Hox* genes, were expressed broadly in the diverse tissue types of the tail, such as muscle, epidermis, and gland cells. Altogether, our work resolves a subset of genes with the earliest expression changes caused by Wnt inhibition, including PCGs and novel genes regionally expressed in other differentiated tissues, and shows that multiple differentiated tissues can dynamically maintain region-appropriate gene expression in response to constitutive Wnt signaling.

## Conservation of Wnt targets in distantly related bilaterians

The ubiquity of a role for Wnt signaling in regulation of AP-axis pattern in animals indicates that understanding the program it regulates is a fundamental problem of developmental biology. We compared Wnt-driven gene expression programs in the posterior of planarians and acoels, an evolutionary comparison that should provide insight into primary axis formation across bilaterians. A small group of genes are expressed in the posterior of both species and respond acutely to changes in Wnt signaling—including the transcription factor-encoding *sp5* gene, *Hox* genes, soluble *guanylate cyclase* genes, and genes encoding Wnt-pathway components.

We also found a large number of Wnt targets that were not commonly Wnt-regulated in planarians and acoels. This raises the possibility that genes involved in species-specific posterior biology can come under the influence of Wnt signaling in the course of evolution. It will be interesting to investigate this possibility to understand the evolution of axial features across phyla.

In addition to being Wnt targets in *Hofstenia* and planarians, additional attributes differentiate *sp5* and *Hox* genes from more canonical planarian PCGs: they are expressed in many differentiated cell types in the tail and their expression changes more rapidly during regeneration than does the expression of many other PCGs. These genes are thus good candidates to be mediators of the regional influence of Wnt signaling in not only muscle and neoblasts, but also in differentiated cells. Indeed, planarian *sp5* was required for regulation of a Wnt target in gland cells.

SP5 belongs to the KLF/SP family of transcription factors and the *sp5* gene is a direct target of Wnt signaling in vertebrates [31,32]. Inhibition of *sp5* along with genes encoding Wnt ligands in mice and zebrafish leads to truncated tail development [51,52]. *sp5* is also Wnt-sensitive in the basal chordate amphioxus [53]. In planarians, *sp5* is expressed in the tail and down-regulated following *β-catenin* RNAi [25]. Most work on the role of SP5 as a transcription factor comes from human and mouse cell culture, demonstrating that SP5 influences Wnt target gene expression. However whether SP5 acts as an activator or repressor of these genes has been unclear [48,49]. Recent work in *Hydra* has shown that SP5 acts to repress a Wnt target to restrict head organizer activity [33]. We found that planarian *sp5* inhibits Wnt-driven expression of trunk genes in the tail to establish trunk-tail expression domains, representing an incoherent feed-forward loop (Fig 9). This suggests that *sp5* represses rather than activates Wnt-dependent gene expression in planarians.

We also found that Wnt-driven expression of *sp5* in the posterior is a feature of acoels, a basal bilaterian. *sp5* RNAi in acoels led to up-regulation of two AP regionally expressed *frizzled* genes by RNA-sequencing and expansion of the expression domain of a novel gene from the trunk into the tail. Future work could aim to determine if these genes are direct SP5 targets and the functional role of these gene expression changes in acoels. These results in *Hofstenia* are similar to our findings in planarians, with *sp5* helping to restrict trunk gene expression from the tail. We suggest that inhibition of Wnt-regulated genes or Wnt-pathway components

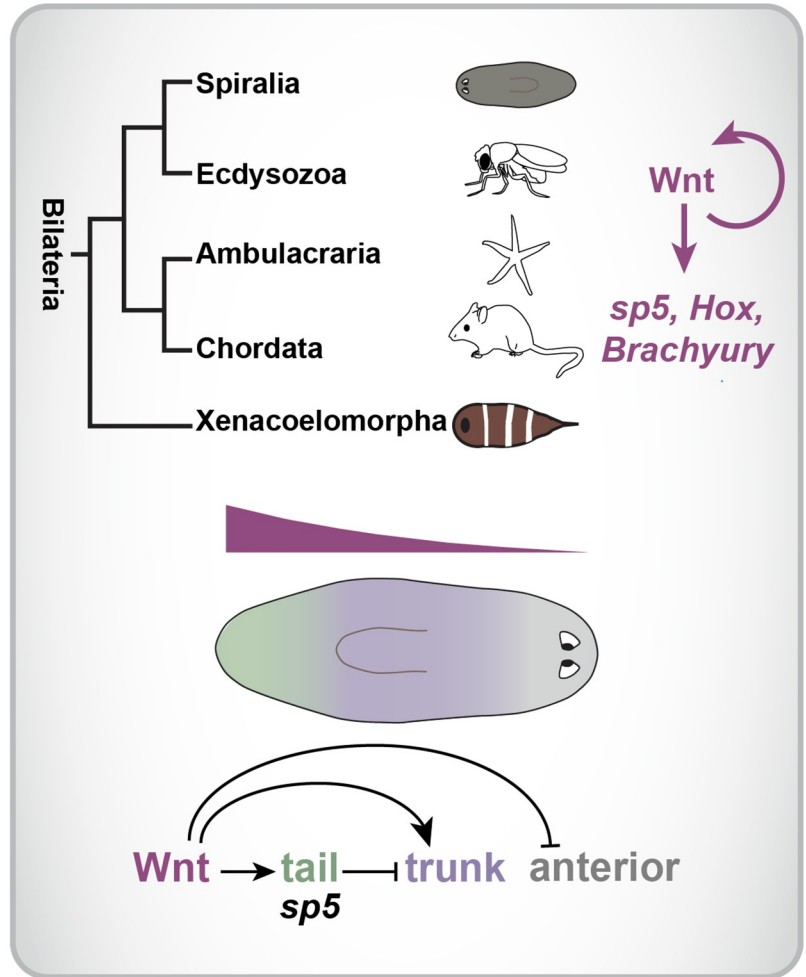

**Fig 9. Model.** We propose that a common Wnt-dependent program including *sp5*, posterior *Hox* genes, *Brachyury*, and Wnt-pathway components was active in the posterior of the bilaterian ancestor. In planarians, *sp5* acts to pattern the primary axis through inhibition of Wnt-dependent trunk gene expression in the tail.

by SP5 in the posterior is a widespread feature of metazoans. This circuit could be utilized to regulate species-specific targets for the establishment and maintenance of diverse tail characteristics.

Regional *Hox* gene expression on the primary body axis is a prominent feature of animals broadly. How such regional expression is regulated across animals is therefore a central problem of developmental biology, yet remains poorly understood. *Hox* regional gene expression domains can also be maintained in adult tissues, such as in adult skin fibroblasts [54,55]. The ubiquity of Wnt signaling in polarizing the AP axis and our finding that *posterior* Hox genes are acutely Wnt-sensitive in planarians and acoels raise the possibility that Wnt signaling regulated the expression of posterior *Hox* genes in the last common ancestor of the Bilateria. We found that planarian *Post-2d* is required for appropriate tail regeneration, suggesting that in addition to their known roles during development, *Hox* genes can be functionally important during regeneration. *Post-2d* was not required for posterior-patterning gene expression during tail regeneration suggesting a more downstream role in the formation of posterior tissues. In

cnidarians, *Hox* genes have been found to regulate expression of Wnts during early axial patterning [56]. In *Hydra*, *Hox1* expression is Wnt dependent [57], but the bilaterian *Hox1* ortholog is an anterior Hox making this comparison at present unclear. Our data are consistent with a possible central role for Wnt in regulating Hox expression domains—a feature that could prove to be a significant patterning concept broadly in animals.

The GUCY1 family of *guanylate cyclase* genes encode soluble receptors for nitric oxide or oxygen. *Hof-gucy1a* encodes an $\alpha$ subunit with a predicted nitric-oxide binding (HNOB) domain whereas planarian *gucy1b2* lacks this domain. This makes it unclear whether these genes would serve similar functions in these species. Future work can aim to understand the role of *guanylate cyclase* genes in the posterior of planarians and acoels, and whether their regulation by Wnt signaling is a widespread feature of bilaterians.

*Brachyury* is a known direct target of Wnt signaling in vertebrates and plays a key role in the formation of posterior mesoderm [58–60,44,45,61]. *Brachyury* expression is Wnt dependent in the posterior of short-germ band insects, such as *Tribolium casteneum*, and in the hypostome of *Hydra* [62–64]. *Brachyury* is also expressed in the blastopore across the Metazoa, and this expression is Wnt-dependent [65,66]. Planarians do not have a detected *Brachyury* ortholog. However, one has been identified in acoels, and is expressed in the mouth and posterior tip. [67]. We found that *Hof-Brachyury* is expressed broadly in the posterior of hatchlings and this expression is Wnt-dependent. This is unlike vertebrates in which *Brachyury* is specifically expressed in the tail bud, and might represent a broader domain of Wnt activity in acoels. This suggests that a Wnt-dependent posterior program including *Brachyury* was active in the posterior of the last common ancestor of the Bilateria.

In vertebrates, the Wnt pathway has multiple roles during different stages of axis development, and could act through distinct targets for mediating the biology of these different phases. For instance, in frogs and fish, before AP axis polarization, β-catenin is required to establish the location of the organizer [68–70]. Later in vertebrate development, Wnt signaling is involved in AP-axis patterning and, subsequently, in posterior growth [6,71]. Posterior patterning followed by an extended period of posterior growth is not overtly recapitulated in posterior regeneration in planarians and acoels, with Wnt signaling having an essentially continuous role during trunk and tail regeneration. How the Wnt-driven posterior programs in planarians and acoels relate to specific Wnt targets in different phases of vertebrate embryonic development will be an important possible direction of study.

Wnt signaling is involved in primary axis patterning throughout the Metazoa, and some Wnt targets (e.g., *sp5* in *Hydra*) predate the emergence of bilaterally symmetric phyla. A small number of select genes are known to regulate primary axis patterning throughout the Bilateria, including *Hox* and *Wnt* genes. We propose that Wnt signaling acts broadly in bilaterians to control posterior expression of *sp5*, posterior *Hox* genes, *Brachyury*, and Wnt-pathway components. Our findings suggest that such a Wnt program was active in the posterior of the last common ancestor of the Bilateria, and will prove to be found broadly in diverse phyla.

## Materials and methods

### Animal care

Asexual *Schmidtea mediterranea* strain (CIW4) were cultured in 1x Montjuic planarian water at 20˚C [72]. Animals were starved 1–2 weeks prior to experiments. *Hofstenia miamia* were maintained at 20˚C in artificial sea water. Hatchlings were fed L-type Brachionus rotifers twice a week. Adults were fed freshly hatched brine shrimp once a week. Two-week old hatchlings were used for experiments [12].

## Cloning

All genes were cloned from cDNA into the pGEM vector (Promega). Cloning primers were designed using the dd_Smed_v6 transcriptome for *Schmidtea mediterranea* (http://planmine. mpi-cbg.de/planmine/begin.do) and the *Hofstenia miamia* transcriptome (Accession: SRP040714). Cloning primers are provided in S8 Table.

## Gene nomenclature

Transcripts with a clear human best BLAST hit were assigned the name of the best BLAST hit in uppercase *italics* (e.g. *BCKHDA*) and the transcript id is provided in parentheses. Transcripts with no clear blast hit are identified by transcript ID.

## Double-stranded RNA synthesis for RNAi experiments

dsRNA was prepared from in vitro transcription reactions (Promega) using PCR-generated forward and reverse templates with flanking T7 promoters (TAATACGACTCACTATAGGG). Each template (16 μl) was mixed with 1.6 μl of 100 mM rNTPs (Promega); 0.6 μl of 1M dithiothreitol (DTT; Promega); 4 μl of T7 polymerase; and 24 μl of 5x Transcription optimized buffer (Promega). Reactions were incubated for 4 h at 37˚C. RNA was purified by ethanol precipitation, and re-suspended in a final volume of 30 μl milliQ $H_2O$. Forward and reverse strands were combined and annealed by heating at 56˚C followed by cooling to 37˚C. Animals used for RNAi were starved at least one week prior to first feeding and animals were fed twice a week. The RNAi food mixture was prepared using 12 μl dsRNA for 30 μl planarian food (homogenized beef liver)[73]. For *wntP-2; sp5* double RNAi experiments, every 1 part *wntP-2* dsRNA was mixed with 2 parts *sp5* dsRNA. *Caenorhabditis elegans unc-22* was used as the control condition [74].

## Fixation (*Schmidtea mediterranea*)

Animals were killed in 5% NAC in PBS for 5 minutes before fixation in 4% formaldehyde for 15 minutes. Fixative was removed and worms were rinsed 2X with PBSTx (PBS + 0.1% Triton X-100). Animals were dehydrated and stored in methanol at -20˚C [75].

## Fixation (*Hofstenia miamia*)

Animals were relaxed in 1% $MgCl_2$ in seawater and subsequently fixed in 4% paraformaldehyde in 1x phosphate-buffered saline with 0.1% Triton-X (PBST) for one hour at room temperature. Animals were dehydrated and stored in methanol at −20˚C [12].

## Whole-mount *in situ* hybridization

RNA probes were synthesized as described previously [76]. Fluorescence *in situ* hybridizations (FISH) for *Schmidtea mediterranea* were performed as previously described [75] with minor modifications. Briefly, fixed animals were bleached, rehydrated and treated with proteinase K (2 μg/ml) in 1xPBSTx. Following overnight hybridizations, samples were washed twice in pre-hyb solution, 1:1 pre-hyb-2X SSC, 2X SSC, 0.2X SSC, PBSTx. Subsequently, blocking was performed in 0.5% Roche Western Blocking reagent and 5% inactivated horse serum in 1xPBSTx. Animals were incubated in antibody overnight at 4˚C. Post-antibody washes and tyramide development were performed as described [75]. Peroxidase inactivation was done in 1% sodium azide for 90 minutes at RT. Specimens were counterstained with DAPI overnight (Sigma, 1 μg/ml in PBSTx). For *Hofstenia miamia*, FISH was performed as described above

with two modifications. First, animals were not bleached. Second, hybridization and pre-hyb solutions were prepared as described in previous work [12].

## Irradiation

Animals were exposed to a 6,000 rads dose of radiation using a dual Gammacell-40 137 cesium source.

## Phylogenetic analysis

*Hofstenia miamia Hox* genes were identified from the transcriptome by reciprocal best BLAST to *Mus musculus* and *Drosophila melanogaster* Hox and ParaHox protein sequences. 9 homeo-domain containing genes obtained by this method were used for phylogenetic analysis. *Schmidtea mediterranea* and *Hofstenia miamia Tbx* genes were identified from the transcriptome by BLAST to *Mus musculus* Tbx protein sequences at a threshold of 1e-06. 5 and 8 genes, respectively, obtained by this method were used for phylogenetic analysis. Multiple sequence alignment of protein sequences was performed using MUSCLE and sequences were trimmed with Gblocks. Phylogenetic trees were constructed using Bayesian inference (MrBayes v3.2.2 x64). Each analysis was performed using two independent runs with four chains each for 1 million generations or more and using a mixture of models for amino acid evolution. Runs converged with an average standard deviation of split frequencies <0.01 and 25% of trees were discarded as burn-in. All protein sequences used for analysis with accession numbers are provided in S6 Table. Nexus files (.nex) for each tree are provided as datasets S1-4.

## RNA sequencing library preparation

Total RNA was isolated using Trizol (Life Technologies) from five to six pooled tails, in biological triplicate. Libraries were prepared using the Kapa Stranded mRNA-Seq Kit Illumina Platform (KapaBiosystems) and sequenced on an Illumina Hi-Seq2000.

## RNA sequencing analysis

Reads were mapped to the dd_Smed_v6 transcriptome (http://planmine.mpi-cbg.de/planmine/begin.do) for *Schmidtea mediterranea* and to the *Hofstenia miamia* transcriptome (Accession: SRP040714) using bowtie-1 [77]. Reads from the same isotig for *S. mediterranea* were summed to generate raw read counts for each transcript. Raw read counts were subjected to independent filtering with the filter criterion, overall sum of counts, to remove genes in the lowest 40% quantile. Differential expression analysis was performed using DEseq [78]. Hierarchical clustering (average linkage, uncentered correlation) was performed using cluster 3.0 with median normalized gene expression data. Heatmaps were generated using pheatmap and are displayed as scaled z scores of gene expression counts. Significance is reported as $p_{adj}$ values.

## Quantitative real-time PCR (qRT-PCR)

Three to four animals were collected per biological replicate with three biological replicates per condition. Total RNA was isolated in 1mL Trizol (Life Technologies, Carlsbad, CA) as per manufacturer's instructions. Samples were triturated using a P1000 tip to homogenize tissue. Following RNA purification and resuspension in MilliQ $H_2O$, concentrations for each sample were determined using the Qubit RNA HS Assay Kit (Life Technologies). 1μg RNA input was used to prepare cDNA with the SuperScript III Reverse Transcriptase kit (Invitrogen). Ct values from three technical replicates were averaged and normalized by the Ct value of the

housekeeping gene *g6pd* to generate ΔCt values. Relative expression levels were determined by the -ΔΔCt method by calculating the difference from the average ΔCt value of control RNAi replicates. Bar graph shows relative expression values as $2^{-\Delta\Delta CT}$ with standard deviation and individual expression values. Statistical tests (one-way ANOVA) were used to determine significance. $p < 0.05$ was used as the significance threshold.

### RNAi injections and soaking

*Hofstenia miamia* dsRNA injections were performed for three consecutive days using a Drummond Nanoject II Auto-nanoliter injector. Needles were pulled from Borosilicate capillaries (#BF100-78-15) on a Sutter Model P-2000 micropipette puller. Animals were injected 2–3 times in one sitting with 32.9nl of dsRNA per injection. RNAi soaking was performed for three consecutive days for 6 hours each. *Hofstenia* were soaked in 12μl dsRNA in 600μl artificial sea water.

### Microscopy and image analysis

Fluorescent images were taken with a Zeiss LSM700 Confocal Microscope. All images are Maximum intensity projections unless otherwise indicated in figure legends. Light images were taken with a Zeiss Discovery Microscope.

### Graphs and statistical analysis

All graphs and statistical analyses were done using the Prism 7.0 software package (GraphPad Inc., La Jolla, CA). Comparisons between the means of two populations were done by a Student's t-test. Comparisons between conditions in a time course was performed by two-way ANOVA. Significance was defined as $p < 0.05$.

### Accession

RNA-sequencing data is available as BioProject PRJNA558985. Gene sequences are available in GenBank with accession numbers MN275808-MN275830, MN305295-MN305319, MN381936-MN381938 and MN400983.

## Supporting information

**S1 Table. Differential expression analysis of control versus *β-catenin-1* RNAi tails in *Schmidtea mediterranea*.**
(XLSX)

**S2 Table. Clusters obtained from hierarchical clustering of genes differentially expressed after *β-catenin-1* RNAi in *Schmidtea mediterranea*.**
(XLSX)

**S3 Table. Differential expression analysis of irradiated and control tails at day 4 after *β-catenin-1* RNAi in *Schmidtea mediterranea*.**
(XLSX)

**S4 Table. Differential expression analysis of control versus *β-catenin-1* RNAi tails in *Hofstenia miamia*.**
(XLSX)

**S5 Table. Clusters obtained from hierarchical clustering of genes differentially expressed after *β-catenin-1* RNAi in *Hofstenia miamia*.**
(XLSX)

**S6 Table. Protein sequences and accession numbers used for phylogenetics analyses.**
(XLSX)

**S7 Table. Differential expression analysis of *postHox* RNAi tails in *Hofstenia miamia* and *sp5* RNAi tails in *Schmidtea mediterranea* and *Hofstenia miamia*.**
(XLSX)

**S8 Table. Primers used for cloning genes from cDNA in *Schmidtea mediterranea* and *Hofstenia miamia*.**
(XLSX)

**S9 Table. List of genes used in this study with accession numbers.**
(XLSX)

**S1 Dataset. Nexus file for Bayesian analysis of SP proteins in *Schmidtea mediterranea* and *Hofstenia miamia*.**
(TXT)

**S2 Dataset. Nexus file for Bayesian analysis of Hox proteins in *Hofstenia miamia*.**
(TXT)

**S3 Dataset. Nexus file for Bayesian analysis of gucy proteins in *Schmidtea mediterranea* and *Hofstenia miamia*.**
(TXT)

**S4 Dataset. Nexus file for Bayesian analysis of Tbx proteins in *Schmidtea mediterranea* and *Hofstenia miamia*.**
(TXT)

**S1 Fig. Expression of genes in the posterior epidermis is down-regulated after *β-catenin-1* RNAi.** (A) Heatmap of genes down-regulated after *β-catenin-1* RNAi at any timepoint (padj<0.05 and log$_2$ fold change ≤-0.5) subjected to hierarchical clustering. Heatmap displays gene expression counts as z scores for time-points post-RNAi feeding. Red box indicates cluster shown in Fig 1B. Differential expression analysis provided in S1 Table. (B) Posterior epidermal expression of *ALDH2* is *β-catenin-1* dependent. FISH for *ALDH2* (magenta) after control and *β-catenin-1* RNAi. Left: Maximum intensity projection of ventral epidermis. Middle: Maximum intensity projection of dorsal gut. Scale bars 200μm. Right: Zoom in of tail tip indicated by yellow box showing *ALDH2* expression in epidermis; *ALDH2* (magenta), DAPI (gray). Scale bars 100μm. (C) Key for tSNE-plots displaying single cells. Major planarian cell types are labelled with distinct colors. (D) *ALDH2* is expressed in many cell types. tSNE-plot of planarian cells colored by *ALDH2* (dd_1339) gene expression (red, high; blue, low). (E) *BCKHDA* is expressed in posterior epidermis. Left: *BCKHDA* expression during homeostasis, 6 days post control and *β-catenin-1* RNAi. Scale bars 200μm. Right top: tSNE-plot of planarian cells colored by *BCKHDA* (dd_10732) gene expression (red, high; blue, low). Right bottom: Zoom in of tail tip indicated by yellow box showing *BCKHDA* expression in dorsal epidermis; *BCKHDA* (magenta), DAPI (gray). Scale bars 100μm. (F) Posterior expression of *gucy1b2* is *β-catenin-1* dependent. Left: *gucy1b2* expression during homeostasis, 6 days post control and *β-catenin-1* RNAi. Scale bars 200μm. Right top: tSNE-plot of planarian cells colored by *gucy1b2* (dd_12650) gene expression (red, high; blue, low). Right bottom: Zoom in of tail tip indicated

by yellow box showing *gucy1b2* expression in posterior epidermis and sub-epidermal cells; *gucy1b2* (magenta), DAPI (gray). Scale bars 100μm.
(TIF)

**S2 Fig. Genes expressed in anterior epidermis and parenchymal cell types are up-regulated in the tail after *β-catenin-1* RNAi.** (A) Heatmap of genes up-regulated after *β-catenin-1* RNAi at any timepoint (padj<0.05 and log$_2$ fold change ≥0.5) subjected to hierarchical clustering. Heatmap displays gene expression counts as z scores for time-points post RNAi feeding. Red box indicates cluster shown in Fig 2B. Differential expression analysis provided in S1 Table. (B) Key for tSNE-plots displaying single cells. Major planarian cell types are labelled with distinct colors. (C) dd_6380 is expressed in anterior epidermis. Top: tSNE-plot of planarian cells colored by dd_6380 gene expression (red, high; blue, low). Bottom: Single dorsal confocal slice showing expression of dd_6380 (magenta) in the epidermis (DAPI, gray). Scale bar 200μm. (D) dd_12049 is expressed in anterior epidermis. Top: tSNE-plot of planarian cells colored by dd_12049 gene expression (red, high; blue, low). Bottom: Single dorsal confocal slice showing expression of dd_12049 (magenta) in the epidermis (DAPI, gray). Scale bar 200μm. (E) dd_11499 is expressed in anterior epidermis. Top: tSNE-plot of planarian cells colored by dd_11499 gene expression (red, high; blue, low). Bottom: Single dorsal confocal slice showing expression of dd_11499 (magenta) in the epidermis (DAPI, gray). Scale bar 200μm. (F) dd_5811 is expressed in parenchymal cells. tSNE-plot of planarian cells colored by dd_5811 gene expression (red, high; blue, low). (G) dd_1379 is expressed in parenchymal cells. tSNE-plot of planarian cells colored by dd_1379 gene expression (red, high; blue, low). (H) *trig-1* is expressed in parenchymal cells. tSNE-plot of planarian cells colored by *trig-1* gene expression (red, high; blue, low). (I) *trig-1* is expressed in marginal adhesive gland cells. Left: FISH for *trig-1* (magenta) and *mag-1* (marginal adhesive gland cells, green) in an uninjured animal. Scale bar 200μm. Right: Zoom of pre-pharyngeal region marked by yellow box. Arrows mark co-expressing cells. Scale bar 20μm.
(TIF)

**S3 Fig. Changes in the expression of patterning genes after Wnt inhibition can occur dynamically in differentiated tissues.** (A) Experimental scheme for RNA sequencing. Animals were subjected to lethal or no irradiation, followed by one RNAi feeding and collection of tails 4 days post feeding. (B) Heatmap of 52 genes identified in Fig 1B at day 4 post RNAi feeding for indicated irradiation conditions. Heatmap displays gene expression counts as z scores. *indicates annotation by best BLAST hit. ***p$_{adj}$<0.05. Differential expression analysis provided in S3 Table. (C) Down-regulation of planarian *sp5* and *Hox* genes after *β-catenin-1* RNAi is irradiation insensitive. FISH for *sp5*, *lox5a*, and *hox4b* after control and *β-catenin-1* RNAi in irradiated animals during homeostasis. Images are presented in grayscale with color inverted. (D) Heatmap of 56 genes identified in Fig 2B at day 4 post RNAi feeding for indicated irradiation conditions. Heatmap displays gene expression counts as z scores. *indicates annotation by best BLAST hit. ***p$_{adj}$<0.05. Differential expression analysis provided in S3 Table. (E) Posterior expansion of anterior expression domains after *β-catenin-1* RNAi is irradiation insensitive. FISH for *wnt2* after control and *β-catenin-1* RNAi in irradiated animals during homeostasis. Arrows indicate posterior boundary of gene expression. Images are presented in grayscale with color inverted. Scale bars, 200μm.
(TIF)

**S4 Fig. Genes differentially expressed along the AP axis are affected after *β-catenin-1* RNAi in *Hofstenia miamia*.** (A) Changes in PCG expression occur by day 6 post *β-catenin-1* RNAi in *Hofstenia miamia*. FISH for anterior PCG *sFRP-1* (green) and posterior PCG *fz-1*

(magenta) at indicated time points post RNAi initiation during homeostasis. Scale bars, 200μm. (B) Heatmap of all genes down-regulated after *β-catenin-1* RNAi ($p_{adj}$<0.1) subjected to hierarchical clustering. Heatmap displays gene expression counts as z scores for time-points post-RNAi feeding. Red box indicates cluster shown in Fig 3B. Differential expression analysis provided in S4 Table. (C) Heatmap of all genes up-regulated after *β-catenin-1* RNAi ($p_{adj}$<0.1) subjected to hierarchical clustering. Heatmap displays gene expression counts as z scores for timepoints post-RNAi feeding. Red box indicates cluster shown in Fig 4B. Differential expression analysis provided in S4 Table. (D) *PKNOX-2* is expressed in anterior epidermis and sub-epidermal cells. FISH for *PKNOX-2* (98052872) (magenta) with DAPI. Single confocal slice of anterior dorsal epidermis, region indicated by yellow box. Scale bar, 50μm. (E) 98052872 is expressed in anterior epidermis. FISH for 98052872 (magenta) with DAPI. Single confocal slice of anterior dorsal epidermis, region indicated by yellow box. Scale bar, 50μm. (TIF)

**S5 Fig. Phylogenetic analysis of SP5 and Hox proteins.** (A) Phylogenetic tree for the placement of *Schmidtea mediterranea* and *Hofstenia miamia* SP5. Bayesian analysis of SP family proteins. Posterior probability indicated at nodes. *Hs (Homo sapiens); Mm (Mus musculus); Dr (Danio rerio); Gg (Gallus gallus); Nv (Nematostella vectenesis)*. Protein sequences provided in S6 Table. Nexus file provided as S1 Dataset. (B) Phylogenetic tree for the placement of *Hofstenia miamia* Hox proteins. Bayesian analysis of Hox and ParaHox proteins. Posterior probability indicated at nodes. *Cap (Capitella teleta); Cs (Cupiennius salei); Bf (Branchiostoma floridae); (Homo sapiens); Dm (Drosophila melanogaster)*. Protein sequences provided in S6 Table. Nexus file provided as S2 Dataset. (C) FISH for *Hof-Hox1*, *Hof-centHox* and *Hof-Cdx* in two-week old hatchlings. Images are presented in grayscale with color inverted. (TIF)

**S6 Fig. Phylogenetic analysis of gucy1 and Tbx proteins.** (A) Phylogenetic tree for the placement of *Schmidtea mediterranea* and *Hofstenia miamia* guanylate cyclase proteins. Bayesian analysis of soluble and transmembrane guanylate cyclase proteins. Posterior probability indicated at nodes. *Hs (Homo sapiens); Mm (Mus musculus); Dm (Drosophila melanogaster); Gg (Gallus gallus); Lm (Limax marginatus)*. Protein sequences provided in S6 Table. Nexus file provided as S3 Dataset. (B) Phylogenetic tree for the placement of *Schmidtea mediterranea* and *Hofstenia miamia* Tbx proteins. Bayesian analysis of Tbx family proteins. Posterior probability indicated at nodes. *Nv (Nematostella vectenesis); Xl (Xenopus laevis); Dr (Danio rerio); Hv (Hydra vulgaris); Dm (Drosophila melanogaster); Cl (Convolutriloba longifissura); Bf (Branchiostoma floridae); Mm (Mus musculus)*. Protein sequences provided in S6 Table. Nexus file provided as S4 Dataset. (TIF)

**S7 Fig. Cell-type gene expression of conserved Wnt targets in the planarian tail.** (A) *sp5* gene expression changes occur by 24 hours post amputation. FISH for *sp5* (magenta) in regenerating heads, trunks, and tails. Yellow arrow indicates anterior boundary of *sp5* expression. (B) t-SNE plot key of cells from the planarian tail. Each major tissue type is labelled with a color. (C) tSNE-plot of planarian tail cells colored by *axinB* gene expression (red, high; blue, low). (D) tSNE-plot of planarian tail cells colored by *fz4-1* gene expression (red, high; blue, low). (E) tSNE-plot of planarian tail cells colored by *fz4-2* gene expression (red, high; blue, low). (F) tSNE-plot of planarian tail cells colored by *gucy1b2* gene expression (red, high; blue, low). (G) Expression of planarian *sp5* and *Hox* genes re-scales prior to *wntP-2* re-scaling in regenerating tail fragments. FISH for *sp5*, *lox5a*, *hox4b*, *Post-2c* (magenta), and *wntP-2* (green)

at 0, 24, and 48 hours post amputation in regenerating tails. Scale bars, 200μm.
(TIF)

**S8 Fig. *sp5* RNAi enhances formation of ectopic mouths in *wntP-2* RNAi animals.** (A) *Post-2d* RNAi leads to tail-regeneration defects in planarians. Live images of regenerating trunk fragments 12 days after amputation. (B) *Post-2d* RNAi head fragments express posterior PCGs during regeneration. FISH for *wnt11-1* and *wnt11-2* pool (magenta) at 5dpa. (C) *Hofstenia postHox* RNAi leads to down-regulation of *Brachyury* in the tail. RNA sequencing on *Hofstenia* tails 7 days after initiation of dsRNA injections for *postHox* or control RNAi. Volcano plot displays genes differentially expressed between control and *postHox* RNAi tails with a $p_{adj} < 0.05$ and $log_2$ fold change $< -1$ or $> 1$, Differential expression analysis provided in S7 Table. (D) Expression of *ndl-3* and *ptk-7* is *β-catenin-1* dependent. FISH for *ndl-3* (magenta) and *ptk-7* (green) 6 days after control and *β-catenin-1* RNAi in homeostasis. (E) qRT-PCR for *wntP-2* expression in 0 hour trunks after 3 weeks of given RNAi conditions. Data is presented as mean ± S.D. ****$p < 0.001$ (F) Plot shows the percentage of RNAi animals that formed ectopic mouths after 6 RNAi feedings. p-values provided for Fisher's exact test. 3 independent RNAi experiments are pooled in this analysis.
(TIF)

## Acknowledgments

We thank the members of the Reddien Lab for comments and discussion.

## Author Contributions

**Conceptualization:** Aneesha G. Tewari, Peter W. Reddien.

**Data curation:** Aneesha G. Tewari.

**Formal analysis:** Aneesha G. Tewari, Peter W. Reddien.

**Funding acquisition:** Peter W. Reddien.

**Investigation:** Aneesha G. Tewari, Jared H. Owen, Christian P. Petersen, Daniel E. Wagner.

**Methodology:** Aneesha G. Tewari, Peter W. Reddien.

**Supervision:** Peter W. Reddien.

**Validation:** Aneesha G. Tewari.

**Visualization:** Aneesha G. Tewari.

**Writing – original draft:** Aneesha G. Tewari, Peter W. Reddien.

**Writing – review & editing:** Aneesha G. Tewari, Jared H. Owen, Christian P. Petersen, Daniel E. Wagner, Peter W. Reddien.

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
