## [Decision Letter · Decision Letter 0]

20 Jul 2019

Dear Dr Reddien,

Thank you very much for submitting your Research Article entitled 'A small set of genes, including SP5 and Hox, are activated by Wnt signaling in the posterior of planarians and acoels' to PLOS Genetics. Your manuscript was fully evaluated at the editorial level and by independent peer reviewers. 

The three reviewers are very positive about your paper but requested a good number of editorial changes and some potential experiments that would strengthen the paper, in particular adding an RNAi experiments for Sp5.

I am sure that you will be able to address all of these comments within the allocated time

We therefore ask you to modify the manuscript according to the review recommendations before we can consider your manuscript for acceptance. Your revisions should address the specific points made by each reviewer.

[LINK]

Yours sincerely,

Claude Desplan

Associate Editor

PLOS Genetics

Kirsten Bomblies

Section Editor: Evolution

PLOS Genetics

**Comments to the Authors:**

Reviewer #1: Wnt signaling plays a vital and highly conserved role in regulating anterior-posterior body patterning across bilaterians. Despite its pivotal role, little is known about the conserved Wnt targets that drive this important developmental process. In this manuscript, Tewari et al., identify the shared Wnt targets responsible for regulating posterior body identity in the planarian Schmidtea mediterranea and the acoel Hofstenia miamia. Specifically, the authors show that Hox genes, the transcription factor sp5, and Wnt signaling components are down-regulated in response to RNAi-mediated b-catenin knockdown in both planarians and acoels. Since acoel worms are Xenacoelomorpha, basal bilaterians separated from planarians by over 550 million years, these newly identified Wnt targets common to both species likely represent a core Wnt-dependent program that is ancestral to all bilaterians. This high-quality work will be of general interest to a broad audience.

Line 307: The authors mention that none of the planarian Hox genes show any observable AP abnormalities after RNAi-mediated knockdown. This group has previously shown that even combinatorial RNAi of three Hox genes, hox4b, post-2c, lox-5a, does not result in AP patterning defect (Scimone et al., 2016). These three genes represent three of the four b-catenin-sensitive Hox genes identified this paper. Does RNAi of all four genes have an effect on AP patterning? Along these lines, does RNAi of the b-catenin target, Hof-Posthox, display any RNAi-mediated AP defects?

Line 312: The images in Figure S5c do not depict Hof-Hox1 and Hof-centHox as being expressed in the “anterior and mid-body” of acoels. Expression seems uniform along the AP axis. Conversely, Hof-Cdx, does not appear to be “expressed broadly on the AP axis of hatchlings” but instead is expressed in a gradient with enrichment in the trunk and tail. The authors should include the expression pattern of the b-catenin-sensitive Hof-postHox.

Line 316: The authors find that planarian and acoel guanylate cyclase homologs “are expressed in the posterior and encode subunits of the GUCY1 family of soluble guanylate cyclase proteins (Fig S6a).” Where is this expression data? Please show in situs in this figure.

What is the evidence that brachyury does not exist in planarians? Can the authors expand on this point in the manuscript? Are the authors referring to work by María Martín-Durán and Rafael Romero (2011) on S. Polychroa? If so, please cite. Otherwise, authors should show orthology of S. mediterranea Tbx genes (place S. mediterranea Tbx genes on the phylogenetic tree in Fig S6) to support their statement.

Reviewer #2: Wnt signaling is a well-known regulator of the posterior of the body during development and maintenance of many animal species. This manuscript explores whether signaling downstream of Wnt is conserved, using planarians and acoels, organisms that—despite many similarities—are only distantly related. Wnt signaling from muscle cells controls maintenance and regeneration of body polarity in both organisms, and Tewari, et al, use this as a jumping off point for molecular studies. The authors use high throughput sequencing to identify genes differentially regulated after downregulation of beta-catenin in planarians and acoels. They then identify genes that the two lists have in common and zero in on sp5, hox family members, and a small number of other conserved genes. The authors go on to make several other important observations. They show that genes differentially expressed in the tail after beta-catenin perturbation are produced in a variety of cell types (not just muscle). They also show that planarian sp5 has a role in setting the boundary that resides between trunk and tail. Finally, the authors show that acoels possess a brachyury homolog for which expression is altered after beta-catenin(RNAi). This manuscript makes a strong contribution to revealing what happens downstream of Wnt signaling in planarian polarity. Further, it supports the idea that a very well-conserved tail signal (Wnt) affects diverse sets of genes in different animal species, including a small common core of targets. This is a thoughtful and compelling paper that should be of interest to readers of PLoS Genetics. I believe that this paper should be considered for publication, especially if the points below are addressed.

1) I recommend altering the title of the manuscript. The title suggests that the number of genes differentially regulated after perturbation of Wnt signaling in the tail is small. In contrast, the authors show many differentially regulated genes, only some of which are conserved.

2) The authors comparative study would be improved by understanding whether acoel genes identified as being downstream of Wnt (e.g. brachyury) affect tail identity. It especially seems important to know whether sp5 affects polarity in the same or different ways in acoels and planarians.

3) The authors emphasize common genes differentially regulated by Wnts in acoels and planarians. One area that was not discussed was the preponderance of genes differentially regulated in the two datasets that were not found to be conserved. I think it would be worth delving into this a bit in the discussion. Is it surprising that so few genes are common? Is it surprising that that the common genes aren’t essential for tail identity (at least not as far as the authors have shown)? Do these observations indicate that a conserved Wnt signal from the tail is handled in a species-specific way almost from step one?

4) Accession numbers for datasets and all transcripts/genes should be deposited and also referenced well in the supplemental material. The information seems to be missing from the manuscript in the methods section and many genes (used for RNAi and ISH) are missing from Table S13. Depositing and referencing sequences of interest (for example in Genbank) will prevent mix-ups in the field and will ensure that the authors receive proper acknowledgement for their work.

Minor points:

1) The timing of the acoel RNAi/RNA-Seq results is later than that of planarians. The authors should describe why they used later time points for this experiment, given that they were focused on identifying the most upstream targets of Wnt signaling. The inclusion criteria for differentially regulated genes also seem to be different and it would be helpful for this decision to be justified in the text.

2) The explanations of clusters in the text got somewhat confusing. For example, line 152 “this set” is unclear. Is this from the 52 genes in the cluster or some other subset? Each of the passages describing clusters should be reviewed for clarity.

3) The sentence in lines 485-6 is unclear. Ubiquitously expressed? Or conserved across all animals?

4) The authors should indicate in the figure legends when the FISH images have been inverted.

5) Figures 3b-c might be clearer if channels were shown separately.

6) In the top panels of Fig. 5b, it isn’t clear which marker is shown.

Reviewer #3: In this work, Tewari et al. combine beta-catenin RNA interference with RNA-sequencing of tail tissues (wherein Wnt activity is high) from planarians and acoels at early time-points post-RNAi treatment. Using this approach, the authors identify a small set of shared genes that are activated by Wnt signaling in the posterior of planarian and acoels, suggesting that a posterior gene expression program is conserved between these organisms. Furthermore, RNAi analysis of sp5 function reveals a role for this gene in the maintenance of proper AP patterning in planarians. This is a solid study and the manuscript is well written. My only major criticism is that the comparative approach taken throughout this study is abandoned for the functional characterization of sp5. The authors should consider performing sp5 RNAi knock down experiments in Hofstenia to test whether there is any obvious phenotype or assess how expression of candidate target genes might be affected as was done for planarians. Analysis of sp5 in acoels or future experiments was omitted from the Discussion.

Additional Comments:

1) P. 11: The header and conclusions in this section alternate between stating a Wnt-driven posterior program is conserved in planarians and acoels (as in the title) and making the bolder claim the program is conserved across the Bilateria. I think it is reasonable to propose the Wnt-driven expression programs observed in the paper (especially in the Discussion) may be broadly conserved, but it does not seem accurate to state the mechanism is conserved in [all] Bilaterians from analysis in two organisms. I suggest editing this section and making statement supported by the data, such as the statements in the model (Fig. 6).

2) The Discussion is a little unfocused. Although interesting, the paragraphs on Hox and guanylate cyclase genes could be omitted. The discussion on the role of Wnt-driven expression in differentiated tissues and the function of sp5 are limited. I would encourage the authors to discuss the broader implications of their findings. Wnt signaling controls homeostatic self-renewal in several adult tissues in mammals. Is there anything known about the role of sp5 or Hox genes in adult tissue homeostasis?

3) I think the authors have space to include some supplementary data in the main manuscript. Figure S4 could be included in the main paper and perhaps S8 could be appended to Figure 5.

Minor Edits:

Line 352: "hour" should read "hours"

Line 422-423: "pattern" should read as "patterns" or "a complex pattern"

Line 441: "neoblast progenitor"; I think the correct terminology is neoblast progeny. Neoblast progenitor would be something that makes neoblasts.

**Have all data underlying the figures and results presented in the manuscript been provided?**

Reviewer #1: Yes

Reviewer #2: No: RNA-Seq data don't have accession numbers and some genes of interest do not have identifiers (Genbank, etc.).

Reviewer #3: Yes

PLOS authors have the option to publish the peer review history of their article (what does this mean?). If published, this will include your full peer review and any attached files.

Reviewer #1: No

Reviewer #2: No

Reviewer #3: No

---

## [Editor Report · Decision Letter 1]

5 Sep 2019

Dear Peter,

We are pleased to inform you that your manuscript entitled "A small set of conserved genes, including sp5 and Hox, are activated by Wnt signaling in the posterior of planarians and acoels" has been editorially accepted for publication in PLOS Genetics. Congratulations for this very beautiful piece of work!

Yours sincerely,

Claude

Claude Desplan

Associate Editor

PLOS Genetics

Kirsten Bomblies

Section Editor: Evolution

PLOS Genetics

**Data Deposition**

http://datadryad.org/submit?journalID=pgenetics&manu=PGENETICS-D-19-00959R1

**Press Queries**

---

## [Editor Report · Acceptance letter]

10 Oct 2019

PGENETICS-D-19-00959R1 

A small set of conserved genes, including sp5 and Hox, are activated by Wnt signaling in the posterior of planarians and acoels 

Dear Dr Reddien, 

We are pleased to inform you that your manuscript entitled "A small set of conserved genes, including sp5 and Hox, are activated by Wnt signaling in the posterior of planarians and acoels" has been formally accepted for publication in PLOS Genetics! Your manuscript is now with our production department and you will be notified of the publication date in due course.

With kind regards,

Kaitlin Butler

PLOS Genetics

On behalf of:
